 **eLIFE**

# Large, long range tensile forces drive convergence during *Xenopus* blastopore closure and body axis elongation

David R Shook[1]*, Eric M Kasprowicz[2], Lance A Davidson[3,4], Raymond Keller[1]

[1]Department of Biology, University of Virginia, Charlottesville, United States; [2]Department of Internal Medicine, Thomas Jefferson University Hospital, Philadelphia, United States; [3]Department of Computational and Systems Biology, University of Pittsburgh, Pittsburgh, United States; [4]Department of Bioengineering, University of Pittsburgh, Pittsburgh, United States

**Abstract** Indirect evidence suggests that blastopore closure during gastrulation of anamniotes, including amphibians such as *Xenopus laevis*, depends on circumblastoporal convergence forces generated by the marginal zone (MZ), but direct evidence is lacking. We show that explanted MZs generate tensile convergence forces up to 1.5 μN during gastrulation and over 4 μN thereafter. These forces are generated by convergent thickening (CT) until the midgastrula and increasingly by convergent extension (CE) thereafter. Explants from ventralized embryos, which lack tissues expressing CE but close their blastopores, produce up to 2 μN of tensile force, showing that CT alone generates forces sufficient to close the blastopore. Uniaxial tensile stress relaxation assays show stiffening of mesodermal and ectodermal tissues around the onset of neurulation, potentially enhancing long-range transmission of convergence forces. These results illuminate the mechanobiology of early vertebrate morphogenic mechanisms, aid interpretation of phenotypes, and give insight into the evolution of blastopore closure mechanisms.

DOI: https://doi.org/10.7554/eLife.26944.001

**\*For correspondence:**
drs6j@virginia.edu

**Competing interests:** The authors declare that no competing interests exist.

## Introduction

Major morphogenic (shape-generating) movements in the development of multicellular organisms occur by integration of local, force-generating activities and force-transmitting properties of individual cells into 'morphogenic machines' that act across the tissue-level length scale. Understanding the physical aspects of tissue movements is essential for understanding how cells and gene products function in morphogenesis (*Keller et al., 2003*; *Keller et al., 2008*), and thus biomechanical measurements, mathematical modeling, and rigorous engineering standards play increasing roles in experimental analyses (see *Jacobson and Gordon, 1976*; *Hardin and Cheng, 1986*; *Priess and Hirsh, 1986*; *Hardin, 1988*; *Hardin and Keller, 1988*; *Koehl, 1990*; *Hutson et al., 2003*; *Keller et al., 2008*; *Rodriguez-Diaz et al., 2008*; *Toyama et al., 2008*; *Layton et al., 2009*; *Varner et al., 2010*). Semi-quantitative biomechanical properties of embryonic cells and tissues have been inferred from responses to micro-dissection (*Beloussov, 1990*; *Fernandez-Gonzalez et al., 2009*; *Ma et al., 2009*; *Solon et al., 2009*; *Martin et al., 2010*; *Fouchard et al., 2011*), and in other cases, direct quantitative measurements have been made (*Adams et al., 1990*; *Moore, 1994*; *Davidson, 1995*; *Davidson et al., 1995*; *Moore et al., 1995b*; *Davidson et al., 1999*; *Zhou et al., 2009*; *Zhou et al., 2010*; *Luu et al., 2011*; *David et al., 2014*; *Feroze et al., 2015*).

A major component of gastrulation in amphibian embryos, such as those of *Xenopus laevis*, in many species of invertebrates and anamniotes, as well as in a few amniotes (see *Stern, 2004*), is 'blastopore closure'. In amphibians, blastopore closure occurs as the ring of presumptive mesoderm

lying at the margin of the blastopore, called the 'Involuting Marginal Zone' (IMZ), rolls, or 'involutes', over the blastoporal lip and simultaneously converges, or decreases in circumference, thereby 'squeezing' the blastopore shut and internalizing the presumptive mesoderm in one motion (*Figure 1*, top panel; *Video 1*, left embryo). Blastopore closure is a systems-level process involving mechanical interplay of a number of regional morphogenic movements in amphibians (*Schechtman, 1942*; *Gerhart and Keller, 1986*; *Keller and Shook, 2004*; *Davidson, 2008*) and other species as well (*Davidson et al., 1995*).

The role of circumblastoporal tensile force in amphibian blastopore closure was inferred from the fact that breaking the continuity of the IMZ transverse to the axis of its convergence, and putative tensile force, results in catastrophic failure of normal involution and blastopore closure (*Schechtman, 1942*; *Keller, 1981*, *1984*); reviewed in *Keller et al., 2003*). The source of this tensile force was thought to be Convergent Extension (CE), an autonomous force-producing process of the IMZ in which its cells express a mediolaterally polarized cell motility, resulting in the intercalation of cells along the mediolateral axis to produce a narrower, longer array, which generates a primary convergence force, and in turn, an extension force (*Moore, 1994*; *Zhou et al., 2015*). Disrupting this mediolaterally polarized cell motility (called 'mediolateral intercalation behavior', MIB) by perturbation of the planar cell polarity (PCP) pathway, blocks convergence in embryos and explants (*Djiane et al., 2000*; *Tada and Smith, 2000*; *Wallingford et al., 2000*; *Habas et al., 2001*; *Goto and Keller, 2002*; *Habas et al., 2003*; *Ewald et al., 2004*). Finally, *Feroze et al., 2015* measured a convergence force of ~0.5 μN at the blastopore lip of Xenopus embryos, and *Zhou et al. (2015)* measured the extension force after blastorpore closure.

Although it was originally thought that CE was the sole source of convergence force in Xenopus, recent work (*Shook et al., 2018*) shows that a second convergence process, Convergent Thickening (CT), is also involved. CT was originally defined in explants of the ventral IMZ of *Xenopus* as an active, radial thickening, perpendicular to the embryonic surface or planar dimension of the tissue, the IMZ in this case, that results in decreasing its planar dimension (convergence), and it was thought to occur only ventrally (*Keller and Danilchik, 1988*). Recently, we found that CT occurs throughout the pre-involution IMZ, not just ventrally, and that the cells expressing CT undergo a transition to express CE at involution (*Shook et al., 2018*) (see *Figure 1*). CT is in many respects the opposite of what is known by convention as radial intercalation (RI), in which cells intercalate between one another perpendicular to the plane of the tissue, which results in the thinning (and spreading) of a tissue. The mechanism involves chemotactic polarization of movement (*Szabó et al., 2016*), integrin-fibronectin signaling (*Marsden and DeSimone, 2001*), and boundary capture (*Keller, 1980*; *Szabó et al., 2016*). During radial intercalation (often resulting in epiboly) cells enter the surface plane of the tissue, whereas CT is the reverse in that cells leave the surface of the tissue, minimizing tissue surface area, resulting in thickening and convergence of the tissue. Coupled with convergence by MIB, RI helps generate extension; in the absence of RI, MIB could instead lead to thickening (see *Keller et al., 2000*). Another major difference between RI and CT is that, like MIB, RI depends on polarized cell behaviors (*Ossipova et al., 2015*), whereas our understanding of CT is that the underlying surface tension based mechanism does not require a polarized cell behavior, only motility to maximize high-affinity cell-cell contacts (*Shook et al., 2018*). In the geometric context of the IMZ, however, the collective result of the cell motility driving CT does result in a polarized, circumblastoporal tension.

Here we use various explants of normal embryos, including the entire IMZ, to measure the composite circumblastorporal convergence forces generated by CT and CE, and we use explants of ventralized embryos lacking the dorsal tissues expressing CE to measure forces generated by CT alone. Therefore it is important to understand at the outset the different spatial and temporal dynamics of CT and CE expression during early development. In our working model, CT (*Figure 1A*, white symbols) occurs throughout the IMZ as gastrulation begins and decreases the circumference of the IMZ symmetrically from all sides, which tends to push it toward the blastoporal lip and contributes to blastopore closure by advancing the blastoporal lips across the vegetal endoderm (*Figure 1A*, green arrows, VE; *Video 1*, left embryo). The early involuting cells of the presumptive head, heart and lateroventral mesoderm (*Figure 1A,B*, orange), which expressed CT in the preinvolution IMZ, transition to migrating directionally across the blastocoel roof toward the animal pole after involution (*Figure 1B*, gray arrows, dorsal cutaway) (*Winklbauer and Nagel, 1991*). In contrast, the following, later-involuting presumptive notochordal and somitic mesodermal cells (*Figure 1A,B*, magenta and

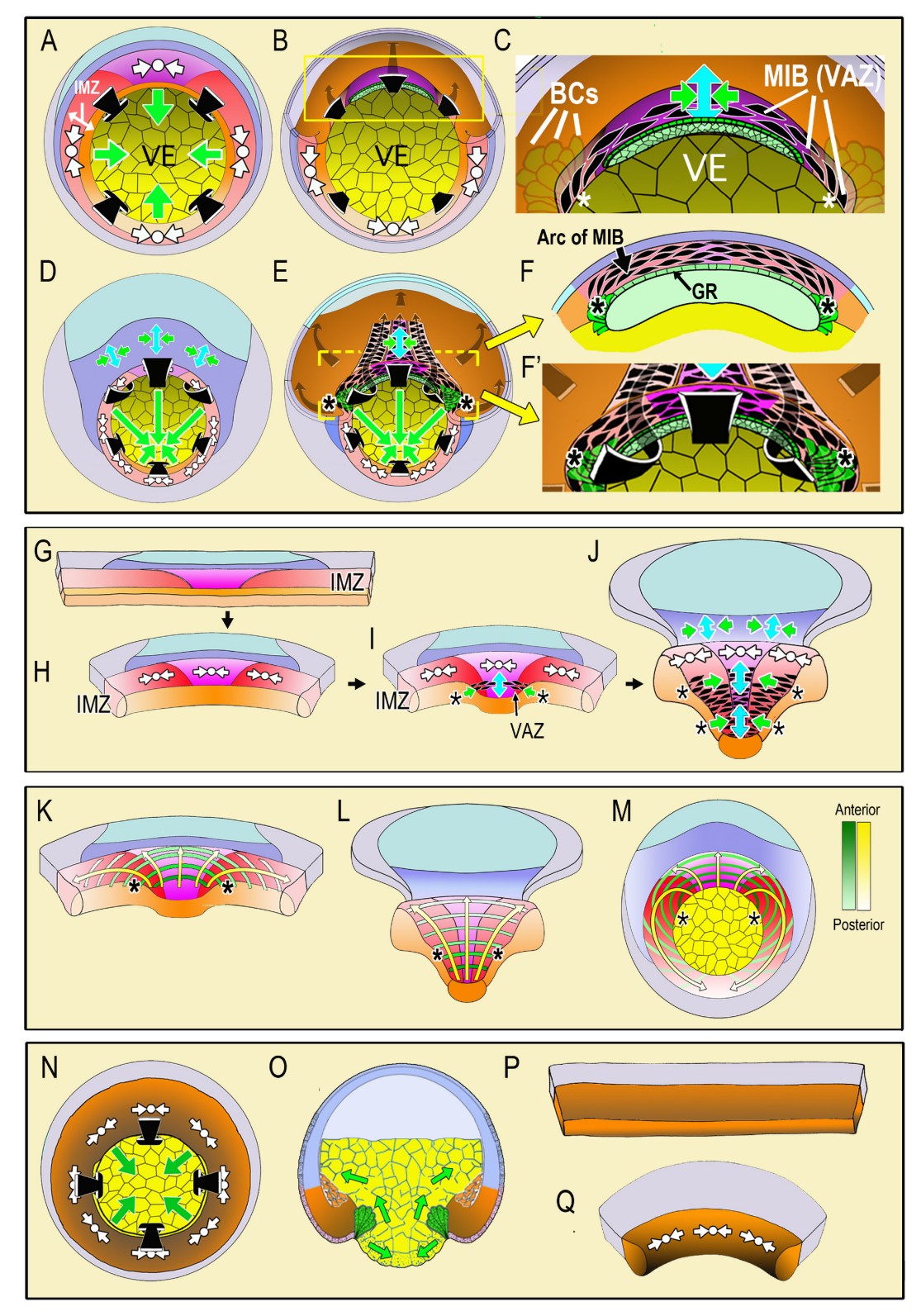

**Figure 1.** Diagrams illustrate the spatial and temporal aspects of the expression of Convergent Thickening (CT) and Convergent Extension (CE) in embryos and explants. In the top panel, a vegetal view at the onset of gastrulation (**A**; *Video 1*, 285 min) shows expression of CT (white symbols, implying circumferential convergence and thickening perpendicular to the surface of the embryo) in the Involuting Marginal Zone (IMZ), and the effect of its convergence in producing involution (black arrows) and blastopore closure (green arrows, (**A**). A cutaway of the dorsal sector of a midgastrula, *Figure 1 continued on next page*

*Figure 1 continued*

showing post-involution IMZ tissues (**B**). An enlargement (**C**) show the onset of CE (green arrows = convergence, blue arrow = the resulting extension). CE is produced by expression of mediolataeral intercalation behavior (MIB) which is expressed initially in the shape of an arc of elongated, intercalating cells (the Vegetal Alignment Zone, VAZ) attached at both ends (asterisks) to the vegetal endoderm (VE) in the region of the bottle cell (BCs). A vegetal view of the late gastrula (**D**; *Video 1*, 525 min) shows continued expression of CT in the IMZ. A cutaway of the same stage (**E**) shows the progressive, posterior expansion of the array of MIB-expressing cells, which advances with the closing blastopore as more cells involute and are added to the array. MIB arcing across the inside of the blastopore (flanked by asterisks) drives CE that acts with CT outside to close the blastopore. Enlargement (**F**) showing a cross-section at the level of the yellow bar in (**E**) showing continued expression of MIB anteriorly, which converges and extends the post-involution notochordal and somitic mesoderm along the length of the axis, which lies between the posterior neural plate and the gastrocoel roof (GR) in embryos. In the middle panel, diagrams of explants of the circumblastoporal region show expression of CT movements at the early gastrula (**G-H**; *Video 2*, 12:26:04 to 13:46:56), the onset of CE (and MIB) to form the VAZ at the midgastrula stage (**I**), and the posterior progression of CE/MIB, as cells expressing CT transition into expressing MIB and CE (**J**; *Video 2*, after 13:46:56). In explants, MIB/CE pulls the unanchored lateral margin of the somitic mesoderm medially while extending and narrowing the somitic and notochordal mesoderm (**I–J**). CT feeds cells into the A-P progressive expression of MIB (**I–J**). Note that MIB occurs in the deep layers of the IMZ, underneath the superficial epithelium, so is not visible in movies of explants shown here, although the resulting CE movements are visible. The bottom panel shows the progressive transition of cells expressing CT to expressing CE and MIB from anterior to posterior (yellow arrows) and the progressive pattern of anterior-to-posterior hoop stress (green hoops) in explants at the midgastrula stage (**K**), the late gastrula (**L**) and the presumptive pattern mapped on the midgastrula embryo (**M**). The last panel shows expression of CT in ventralized embryos, which lack presumptive somitic, notochordal and neural tissue, and thus lacking CE/MIB, and express only CT, which closes the blastopore symmetrically (**N**-vegetal view; **O**-sectional view). Explants from such embryos show only CT (**P-Q**, *Video 3*). Presumptive tissues are indicated (orange- head, heart, lateroventral mesoderm; magenta- notochord; red- somitic mesoderm; dark blue-posterior neural, hindbrain, spinal cord; light blue-forebrain; gray-epidermis; yellow- vegetal endoderm). Shading from dark to light, where used, indicates progressively more anterior to posterior position, respectively.

DOI: https://doi.org/10.7554/eLife.26944.002

red, respectively), which also expressed CT in the preinvolution IMZ, transition on involution to expressing MIB, and thus CE (*Figure 1C*, enlargement of 1B, MIB – aligned, black, fusiform cells; CE- green/blue arrows symbol). Expression of MIB is progressive, beginning anteriorly at the midgastrula (stage 10.5) with the formation of an arc of intercalating cells just inside the dorsal blastopore, called the 'Vegetal Alignment Zone' (*Figure 1C*, VAZ). The VAZ spans the dorsal aspect of the blastopore and is anchored at both ends to the vegetal endoderm near the bottle cells (*Figure 1C*, asterisks indicating anchorage point, VE, BCs,). As preinvolution IMZ cells continue to express CT and aid in blastopore closure throughout gastrulation (*Figure 1D*), MIB progresses posteriorly from its anterior origin in the VAZ. As preinvolution cells undergo involution, they cease expressing CT and begin expressing MIB as they round the blastoporal lip (*Figure 1E*), thereby adding to the posterior end of the postinvolution array of MIB expressing cells that span the roof of the gastrocoel (*Figure 1E,F*). The array of postinvolution, MIB expressing cells increases in length and number of cells acting in parallel to drive CE (green-blue arrows symbol, *Figure 1E*). The cells that are progressively added at the posterior end of this array as the blastopore closes contribute to a constricting arc or hoop of tension just inside the blastoporal lip (*Figure 1E*), just as those that formed the VAZ did initially (*Figure 1C*). The posterior progressivity of expression of MIB/CE results in an increasingly anisotropic blastoporal closure from St. 10.5 (midgastrula) onward, with the dorsal/dorsolateral sides closing over the vegetal endoderm faster than the ventral/ventrolateral sides (green arrows, *Figure 1D,E*). These events are based on studies in embryos (*Keller, 1981*; *Keller, 1984*; *Keller and Danilchik, 1988*; *Keller et al., 1989*; *Lane and Keller, 1997*), correlation with live imaging of MIB in explants of the marginal zone (*Wilson and Keller, 1991*; *Keller and Winklbauer, 1992*; *Shih and Keller, 1992b*; *Shih and Keller, 1992a*; *Domingo and Keller, 1995*), and characterization of CT (*Shook et al., 2018*).

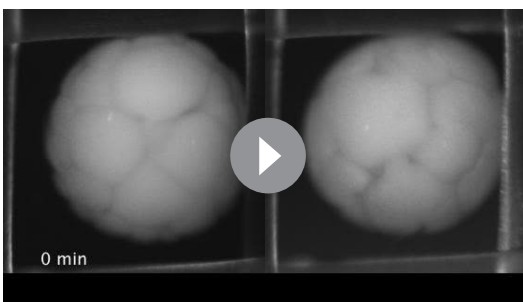

**Video 1.** Movie comparing gastrulation and BP closure in normal (left) and ventralized (right) embryos. Timestamp shows minutes elapsed. The movie begins during cleavage stages and runs through neurulation. Gastrulation begins at control stage 10 (G0) at 285 min, stage 10.5 (G+2hr) at 405 min.

DOI: https://doi.org/10.7554/eLife.26944.003

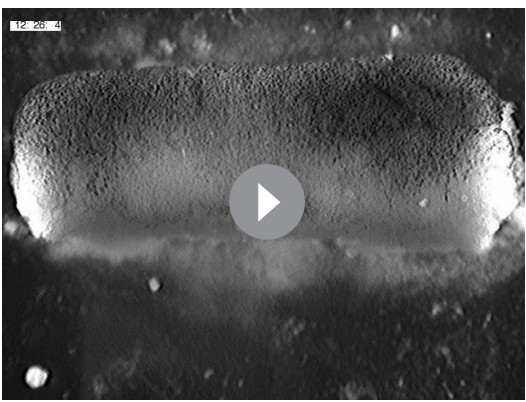

**Video 2.** Movie showing unencumbered giant sandwich explant. Timestamp shows hh:mm:ss. Stage 10.5 (G + 2 hr) at about 13:34:57.
DOI: https://doi.org/10.7554/eLife.26944.004

To assay circumblastoporal tensile forces, we use 'giant sandwich explants' consisting of the entire IMZ plus posterior ectodermal and neural tissues (*Figure 1G*). These explants reproduce the major movements described above in embryos, excepting involution (*Video 2*). CT occurs early and symmetrically across the entire mediolateral extent of the IMZ, beginning at the onset of gastrulation, and produces a progressive thickening and convergence of the IMZ (*Figure 1G–H*, white symbols; *Video 2*) (*Shook et al., 2018*), a convergence that we believe acts as a preloading force that pushes the IMZ toward the blastoporal lip and tends to close the blastopore in embryos as described above. In explants, although IMZ cells do not involute, they nevertheless progressively stop expressing CT and express *postinvolution behaviors*, including MIB-driven cell intercalation (resulting in CE) initiated in the form of the VAZ (*Figure 1I*, black, fusiform shapes), beginning at the midgastrula stage. Here, in the explant, the ends of the VAZ are unanchored, ending within the vegetal edge of the explant (*Figure 1I*, asterisks) and when MIB-driven cell intercalation shortens the arc, it acts with CT in the more posterior portions of the IMZ to converge the IMZ medially. As MIB progresses, the explant begins to undergo CE (*Figure 1I*, green/blue arrows). As MIB spreads posteriorly, it incorporates more of the cells in the IMZ in CE movements, progressively narrowing and elongating the axial and paraxial mesoderm from anterior to posterior (*Figure 1J*, green/blue arrows; *Video 2*), as it would in the embryo (*Figure 1E*). The posterior advance of MIB (*Figure 1J*, black fusiform shapes flanked by asterisks) marks the transition zone between the thick zone of continuing CT and the narrowing and extending zone of CE, and corresponds to the point at which cells would be involuting in an intact embryo.

The presumptive pattern of the future, post-involution expression of MIB (*Figure 1E,I,J*) is mapped on to the IMZ of the giant explant at early and late stages (*Figure 1K,L*) and on the IMZ of the whole embryo (*Figure 1M*) to illustrate the A-P progression of MIB driven arc-shortening that drives CE. This pattern of MIB expression was determined from time-lapse imaging of the progress of MIB across open-faced explants (*Shih and Keller, 1992b*; *Domingo and Keller, 1995*; *Keller et al., 2000*). Congruent with the post-involution CE of the mesoderm in the whole embryo, the overlying posterior neural tissue (spinal cord/hindbrain) also undergoes CE (*Figure 1D*, dark blue region), a CE that also occurs in explants (*Figure 1D,J*) (*Elul et al., 1997*; *Davidson and Keller, 1999*; *Elul and Keller, 2000*; *Ezin et al., 2003*; *Ezin et al., 2006*; *Rolo et al., 2009*; *Ossipova et al., 2014*). MIB occurs in various forms in ascidians (*Munro and Odell, 2002*), nematodes (*Williams-Masson et al., 1998*), flies (*Irvine and Wieschaus, 1994*; *Bertet et al., 2004*), fish (*Jessen et al., 2002*; *Glickman et al., 2003*; *Lin et al., 2005*), and in the mesoderm (*Yen et al., 2009*) and neural tissue (*Williams et al., 2014*) of the mouse.

In order to study and measure CT alone, we use giant explants of ventralized *Xenopus* embryos, which lack notochordal, somitic, and neural tissues and therefore do not express CE. Such embryos nevertheless involute their IMZ and close their blastopores (*Scharf and Gerhart, 1980*) (*Video 1*, right embryo), suggesting that CT alone can close the blastopore (*Figure 1N,O*)(see *Keller and Shook, 2004*). IMZ explants from ventralized embryos show a rapid, near uniform CT throughout the IMZ (*Figure 1P,Q*, *Video 3*) (*Shook et al., 2018*).

We assayed forces generated by explants including some or all of the IMZ, expressing CT and CE, together or alone, using a mechanical measuring device (the 'tractor pull' apparatus; see Methods, *Figure 2*). We show that the IMZ can generate and maintain large, constant, convergence forces and transmit them over long distances for long periods. We also use a uni-axial tensile stress-relaxation test (*Wiebe and Brodland, 2005*; *Benko and Brodland, 2007*; *Harris et al., 2012*; *Harris et al., 2013*) to measure stiffness of embryonic tissues, which defines their deformation (or 'strain') in response to stress (force per unit area). We also demonstrate a previously unknown

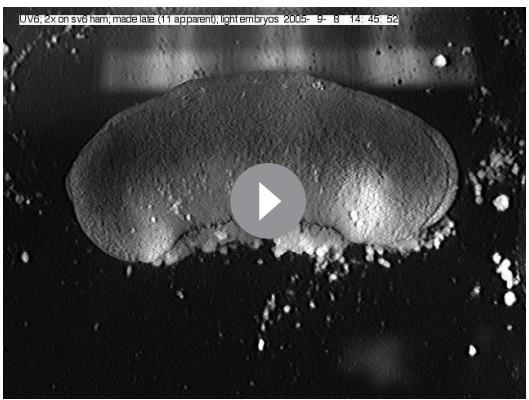

**Video 3.** Movie showing ventralized giant sandwich explant. Timestamp shows hh:mm:ss
DOI: https://doi.org/10.7554/eLife.26944.005

transition in which cells generate most or all of the force around the blastopore by CT until midgastrulation, when cells begin to progressively involute and generate force via the expression of MIB. These methods provide a quantitative approach for evaluating the cellular and molecular mechanisms of developing convergence (tissue shortening) forces and tensile stiffness. Quantification of tensile convergence force and tensile stiffness offers new insights into the causes of failure of blastopore closure, body axis extension, and neural tube closure. Failures of these processes produce common, often linked, but poorly understood phenotypes, which result from genetic and molecular lesions, notably of the PCP pathway (*Ewald et al., 2004*), and are of biomedical importance in neural tube defects.

## Results

### Giant sandwich explants recapitulate most of the in vivo convergence movements of the IMZ

Time-lapse movies show that the mesodermal (IMZ) and neural (NIMZ) tissues in unencumbered (unrestrained) giant sandwich explants (e.g. *Video 2*) undergo convergence similar to that seen in embryos (e.g. *Video 1*), except that the rate peaked earlier in explants (*Figure 2—figure supplement 1*) and was only 57% of that in whole embryos during gastrulation (*Table 1*). Imaging explants as they were made revealed 600 %/hr convergence in the first three minutes after cutting (*Figure 2—figure supplement 2*), two orders of magnitude faster than that of post-construction explants or embryos (*Table 1*), suggesting that convergence against resisting tissues (e.g. the vegetal endoderm) resulted in stored elastic energy in the embryo, in line with prior findings (*Beloussov et al., 1975*; *Beloussov, 1990*; *Fung, 1993*). Therefore rapid, unmeasured convergence of the IMZ occurred when freed of this resistance at explantation. Unencumbered giants and intact embryos reached a minimal rate of convergence by 9 hr after the onset of gastrulation (G+9hr) (*Figure 2—figure supplement 1*), when involution is complete and convergence occurs only as CE of the involuted mesodermal tissues and overlying neural tissue. The dorsal tissues in giant sandwiches converged and extended well (*Figure 2—figure supplement 3*), and tissue differentiation was as expected from previous work, as assayed by markers for notochord and somitic mesoderm (*Figure 2—figure supplement 4A–H*) (see *Keller and Danilchik, 1988*; *Poznanski et al., 1997*).

### Giant sandwich explants generate a consistent pattern of force during blastopore closure

The generation of force by IMZ tissues lying around the blastopore (circumblastoporally) was assayed by cutting these tissues from the embryo and making stable constructs (explants) with plastic attachment strips. As this tissue converged mediolaterally, the tension generated was measured by observing the deflection of a probe pulled by a cleat on one of the attachment strips (*Figure 2*, *Video 4*). Because we knew the spring constant (force per unit tip displacement) of the probe, we could calculate the amount of tension generated by the explant (*Figure 3*), tension that would be expressed circumblastoporally in the intact embryo.

Explants for Giant sandwiches were made by cutting the embryo ventrally and trimming off most of the vegetal endoderm and some of the animal cap region (*Figure 2A,B*). Two Fibronectin coated plastic strips were then inserted between two such explants and the explants allowed to adhere to the strips and heal together within the test chamber (*Figure 2H*). One of the strips was adhered to the substrate (the 'anchor'), while the other was free to move (the 'sled'). The chamber was then positioned on a compound microscope, and an optical fiber probe was positioned adjacent to a 'cleat' attached to the sled (*Figure 2I*). The explant was imaged from above at low magnification,

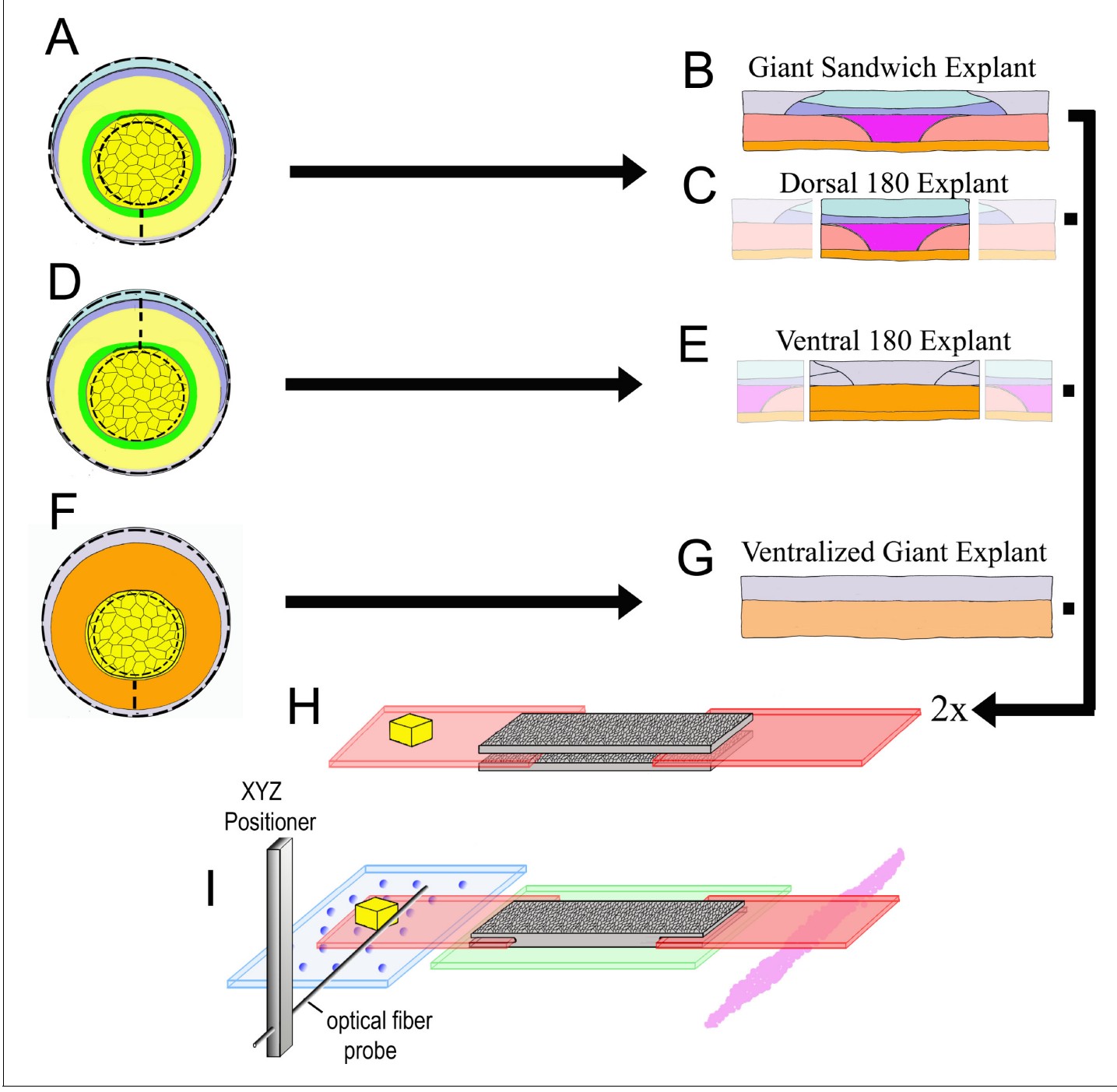

**Figure 2.** Explant construction. Giant sandwich explants are made by cutting early gastrula stage embryos mid-ventrally, then vegetally just below the lower edge of the IMZ, such that the bottle cells are included, then animally roughly 30° above the equator (dashed lines, A; see also *Figure 2—figure supplement 2*). Two such explants are then recombined, inner face to inner face, to make a Giant Sandwich explant (**B, H**). Giant explants contain presumptive notochordal mesoderm (magenta), somitic mesoderm (red), posterior neural tissue (hindbrain-spinal cord), as well as presumptive brain (light blue), epidermis (grey), and migratory leading edge mesoderm (orange). Dorsal 180° explants are made the same way as standard giant explants, with the right and left quarters cut off (**C**). Ventral 180° sandwich explants are constructed similarly, except the IMZ is cut dorsally rather than ventrally (dashed lines, (**D**)). Ventralized giant explants are made from UV irradiated embryos, and thus they form no or very limited dorsal tissues (**F, G**). For mechanical measurements with the tractor-pull apparatus, the two halves of the sandwich are apposed with their inner, deep surfaces next to one another, with fibronectin coated plastic strips, one bearing a raised cleat, inserted at each end (**H**). The explant is allowed to heal and attach to the strips, and then positioned above a cover slip window in a culture chamber (**I**). The stationary 'anchor' strip is attached to the window with silicone high vacuum grease (magenta), and the explant is placed over an agarose pad (green). The moveable 'sled' strip rests on glass beads resting on a cover slip

*Figure 2 continued on next page*

*Figure 2 continued*

filler layer (blue). An XYZ positioner is used to move a calibrated optical fiber probe, mounted on an aluminum bar, near the cleat, and the imaging chamber, which rests on a motorized stage, is then moved such that the cleat is as close to the probe tip as possible without deflecting the probe (See *Video 4*).

DOI: https://doi.org/10.7554/eLife.26944.007

The following source data and figure supplements are available for figure 2:

**Figure supplement 1.** Comparison of convergence along the limit of involution (LI).

DOI: https://doi.org/10.7554/eLife.26944.008

**Figure supplement 1—source data 1.** Source data for convergence rates.

DOI: https://doi.org/10.7554/eLife.26944.009

**Figure supplement 2.** Convergence during explant construction.

DOI: https://doi.org/10.7554/eLife.26944.010

**Figure supplement 2—source data 1.** Source data for Convergence during explant construction.

DOI: https://doi.org/10.7554/eLife.26944.011

**Figure supplement 3.** Comparison of morphogenesis in embryos, giants and tractor pulls.

DOI: https://doi.org/10.7554/eLife.26944.012

**Figure supplement 4.** Notochordal and somitic tissue in tractor pull explants.

DOI: https://doi.org/10.7554/eLife.26944.013

while the probe position was imaged from below at high magnification (see Methods for further details).

Assays of standard giant sandwich explants (*Figure 2B*), beginning between control stages 10.25 (G+1hr) and 11.5 (G+3.5hr), showed a consistent pattern of circumblastoporal (mediolateral)

**Table 1.** Convergence and strain.

Negative strains indicate convergence. 'Standard Pull' refers to standard giant explants that have developed tension within the tractor pull apparatus.

| | Average rate of convergence (µm/min) (n) | Average rate of convergence (%/hr) (n) (SEM) | Strain of dorsal tissue (%/hr) (n) (SEM) | Strain of LV tissue (%/hr) (n) (SEM) | Shear w.r.t. attachment strips (%/hr) |
|---|---|---|---|---|---|
| Intact embryo, LI, 2 to 7 hr | 10 (2 to 7) | 17.5 (2 to 7) (1.8) | | | |
| Giant explant, unencumbered, LI, 0 to 7 hr | 5 (3 to 10) | 10 (3 to 10) (1.5) | | | |
| Standard pull, probe 3 (2 to 7.5 hr) | | 4.1 (4 to 5) (0.7) | −7.2 (6) (1.5) | 1.1 (6) (1.5) | 3.7 |
| Standard pull, probe 3 (7.5 to 10.5 hr) | | 2.5 (5) (0.8) | −7.0 (6) (2.1) | 1.7 (6) (1.9) | 2.5 |
| Standard pull, probe 3 (10.5 to 15.5) | | 3.7 (5) (0.7) | −5.5 (6) (1.6) | −0.5 (6) (1.3) | 3.2 |
| Standard pull, probe 4 (2.5 to 7.5 hr) | | 4.5 (2 to 4) (0.7) | | | 3.1 |
| Standard pull, probe 4 (7.5 to 10.5 hr) | | 1.7 (4) (0.8) | | | 1.7 |
| Standard pull, probe 4 (10.5 to 16.5) | | 3.0 (4) (0.6) | | | 2.1 |

DOI: https://doi.org/10.7554/eLife.26944.014

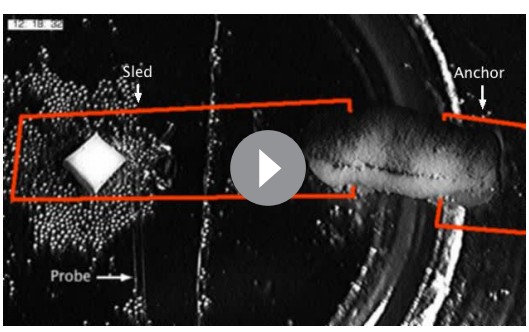

**Video 4.** Movie showing standard giant in tractor pull apparatus. Probe, sled and anchor are indicated in the first frame. Timestamp shows hh:mm:ss. Movie begins shortly after st. 10.5 (G + 2 hr)
DOI: https://doi.org/10.7554/eLife.26944.006

tensile force increasing with time, in two major phases (*Figure 3A*, blue and green lines), using either a stiffer probe (#3, spring constant = 27 μN/μm) or a less stiff probe (#4, spring constant = 12 μN/μm). In the first phase, beginning when the explant pulled the cleat against the probe (*Figure 2I*), usually within the first 3 to 30 min of the assay, force increased steadily to over 1 μN by the end of gastrulation in control embryos (stage 13, G+6hr) and increased further by early neurulation (stage 14, G+7.5hr) when it plateaued at about 2 μN (*Figure 3A*, blue line). A second phase of force increase began 3 hr later (late neurula stage 18, G+10.5hr) with most samples exhibiting a second plateau at about 4 to 5 μN by G+15 to 18 (*Figure 3A*, blue line; *Figure 3—figure supplement 1A*). Probe #3 measurements were similar to those with probe #4 (*Figure 3*, green line; *Figure 3—figure supplement 1B*); they begin earlier only because explant construction was completed earlier. Animal cap sandwiches, which do not normally converge or extend, showed no convergence force (*Figure 3A*, brown line; *Figure 3—figure supplement 1C*), thus ruling out healing and other artifacts. Probe drift and friction were accounted for (see Appendix 1; *Figure 3—figure supplement 2*,*3*).

Immunohistochemical staining showed normal differentiation of somitic and notochordal tissues undergoing CE in mechanically loaded (encumbered) sandwiches (Appendix 2; *Figure 3—figure supplement 4A-H*). As in unencumbered sandwiches, tissues in each half of the sandwich were fused with their counterpart in the other half (see *Keller and Danilchik, 1988*; *Poznanski et al., 1997*). Notochords were sometimes split posteriorly (*Figure 2—figure supplement 4C,G*), perhaps related to retarded convergence compared to that of unencumbered giants (*Figure 2—figure supplements 1B* and *3*). Also the NIMZ, especially the non-neural portion, converged very little in the tractor pull, as compared to unencumbered giants (*Figure 2—figure supplement 1B* and *3*; *Videos 2* and *4*).

## Contributions of the changing expression of CT and CE to convergence force

### CT generates convergence force early and throughout gastrulation.

Using standard giant explants, and dorsal, ventral, and ventralized explants allowed us to ferret out the relative contributions of CT and CE during the progressive transition from CT to CE. Giant explants express CT alone in early gastrula stages, followed by progressive transition to expression of MIB/CE. The IMZ of unencumbered giant explants converges equally across its mediolateral extent, without anisotropic (dorsally-biased) extension through G+2hr (stage 10.5) (*Figure 1G,H*; *Video 2*, through about 13:34:57). Thus CT is expressed early, from the onset of gastrulation, and everywhere in the IMZ, rather than just ventrally and later, as previously thought (*Keller and Danilchik, 1988*). To measure early forces, giant explants were made from late blastulae (the future dorsal side identified by 'tipping and marking'; see Methods), mounted in the apparatus and measured before gastrulation began and prior to expression of MIB/CE. Tension appeared as early as stage 10 (*Figure 3B*, light blue line; *Figure 3—figure supplement 1D*) and rose to 0.3 μN of force prior to the onset of MIB, which occurs at G+2hr (*Shih and Keller, 1992b*; *Lane and Keller, 1997*), demonstrating that CT alone generates this early force. Ventral 180° sandwich explants (*Figure 2D,E*) and explants of the entire marginal zone of UV ventralized embryos (*Figure 2F,G*), both of which largely lack dorsal, CE expressing tissues (see Appendix 2; *Figure 2—figure supplement 4M–P*), converge equally across their mediolateral extent until reaching an equilibrium state of convergence and thickening (*Video 3*). These ventral sandwich explants showed initial force increase similar to standard giant explants and plateaued at about 2 μN, similar to standard giant explants, but neither exhibited the second phase of force production (*Figure 3D*, yellow, orange lines; *Figure 3—figure supplement 1G,H*). These results show that during gastrulation, forces equivalent to those

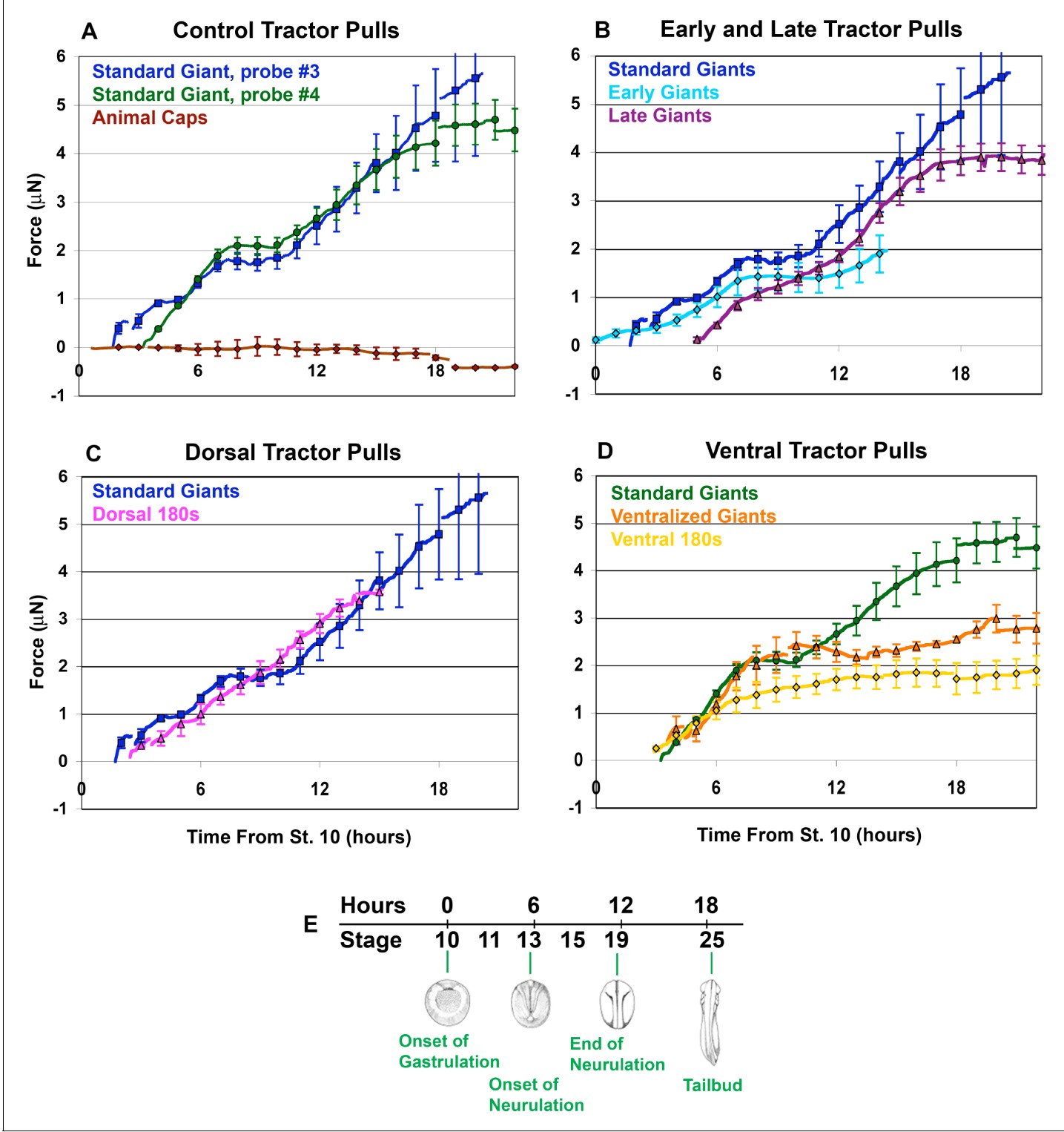

**Figure 3.** Force vs. time traces for tractor pulls. Mean force production over time is indicated (**A-D**; solid lines). Time is measured from the onset of gastrulation at stage 10, and the correspondence with developmental stage is shown (**E**). Hourly means are shown as symbols, with standard errors of the hourly means indicated. The onset of individual traces represents the time at which the sled was initially pulled against the probe, with the exception of the Animal Cap explants. All pulls were against probe #3, except 'Standard Giants, probe #4' (green, **A and D**) and the 'Ventralized Giants' and 'Ventral 180 s' (orange and yellow respectively, (**D**). The force trace for Standard Giant explants vs. probe #3 (blue) is included for all graphs except where only probe #4 was used (**D**). Gaps in force traces represent points at which different numbers of explants are included in the mean force trace. Controls (**A**) included Standard Giant sandwich explants (vs. probe #3, dark blue, n = 2 up to 2 hr, n = 4 through 15 hr; vs. probe #4, green, n = 6

*Figure 3 continued on next page*

eLIFE Research article

Developmental Biology and Stem Cells | Structural Biology and Molecular Biophysics

*Figure 3 continued*

through 12 hr, n = 5 through 18 hr), Animal Cap sandwich explants (purple, n = 4 from 5 to 17 hr, 2 to 3 otherwise). Early and Late tractor pulls (**B**) include Early Giant explants (from stage 10; light blue, n = 3) and Late Giant explants (from stage 12.5; purple, n = 5). Dorsal tractor pulls (**C**) include Dorsal 180° explants (pink, n = 3). Ventral tractor pulls (**D**) include UV ventralized explants (orange, n = 3 at 4 hr, 4 from 5 to 7 hr, 5 from 8 to 9 hr and 4 from 10 to 20 hr), ventral 180° explants (yellow, n = 3 at 3 hr, 4 from 4 to 20 hr) and are compared to Standard Giant explants vs. probe #4.
DOI: https://doi.org/10.7554/eLife.26944.015

The following source data and figure supplements are available for figure 3:

**Figure supplement 1.** Plots of all the individual force traces.
DOI: https://doi.org/10.7554/eLife.26944.016

**Figure supplement 1—source data 1.** Source data for all individual force traces; data for each panel (**A–H**) broken down by sheets within the Excel file.
DOI: https://doi.org/10.7554/eLife.26944.017

**Figure supplement 2.** Probe drift tests.
DOI: https://doi.org/10.7554/eLife.26944.018

**Figure supplement 2—source data 1.** Source data for probe drift tests; data for each panel (**A–C**) broken down by sheets within the Excel file.
DOI: https://doi.org/10.7554/eLife.26944.019

**Figure supplement 3.** Sled friction tests.
DOI: https://doi.org/10.7554/eLife.26944.020

**Figure supplement 3—source data 1.** Source data for sled friction tests.
DOI: https://doi.org/10.7554/eLife.26944.021

**Figure supplement 4.** Response of explants in tractor pull after tension increase or relaxation.
DOI: https://doi.org/10.7554/eLife.26944.022

**Figure supplement 4—source data 1.** Source data for Increased tension or Relaxation during tractor pulls, panels A,B and F,G on different sheets within Excel file.
DOI: https://doi.org/10.7554/eLife.26944.023

generated by a combination of CE and CT in standard giants, are generated by CT alone in ventral/ventralized tissues and that CT continues to operate if not replaced by CE.

## Explants expressing CE over a larger proportion of their mediolateral aspect have a reduced or no plateau.

Late giant explants, made as giant sandwich explants (*Figure 2A,B*) but remain unencumbered until the late gastrula stage, and Dorsal 180 explants (*Figure 2A,C*), which develop largely dorsal tissues, show a more continuous increase of tensile force, with a moderate or no decline in the rate of force increase (*Figure 3B and C*, magenta and pink lines, respectively; *Figure 3—figure supplement 1E, F*) during the plateau of normal giants, and they show a final plateau of 3.5 to 4 μN (G+14hr; stage 21–22), a little lower and earlier than normal giant explants. This more continuous rise in force is correlated with absent or reduced ventral/ventrolateral regions lying between dorsal MIB/CE expressing tissue and the attachment strips, which suggests that the presence of ventrolateral (presumptive posterior) tissues accounts for the plateau.

## The plateau is likely due to strain in the ventrolateral tissues

Unencumbered giants converge (=negative strain) uniformly along their mediolateral axis (data not shown; see *Video 2*) at 10 %/hr (*Table 1*), whereas encumbered giants converge at 70% of this rate in the dorsal sectors that are expressing CE, but there is no convergence in the more ventrolateral sectors not yet expressing CE during the first phase of force increase (*Table 1*; *Video 4*). Comparison of the rate of shear of the explant with respect to the attachment strips (the rate of convergence of the edges of the widest part of the IMZ minus the convergence of the sled toward the anchor (Appendix 3)), showed that shear accounted for substantially more of explant convergence than did sled movement (*Table 1*). During the plateau, overall IMZ convergence dropped by 1.6 %/hr for probe three while shear dropped by 1.2 %/hr, such that it matched the remaining convergence of 2.5 %/hr (*Table 1*; Appendix 3). By region, the overall decline in convergence was explained by a 0.6 %/hr increase in strain in ventrolateral tissues and a 0.2 %/hr decline in convergence in dorsal tissues. Thus, the plateau is due primarily to the increased strain of the ventrolateral tissues; this could be explained either by a decline in stiffness, or by a decline or stall of force generation in this region.

Also, the end of the plateau in giant explant assays is associated with significantly increased convergence of the ventrolateral regions by 2.2 %/hr and significantly decreased convergence by 1.5 %/hr in dorsal regions (*Table 1*), suggesting an increase in stiffness or force generation in these ventrolateral regions contributes to the end of the plateau.

## Tension developed by explants represents a progressively increasing, instantaneous stall force

Forces generated by the cells within the tissue increase tension and drive convergence. Because the probe resists this convergence, tension across the explant increases over time. Tension increase is limited by the stall force of the motors involved (CT, CE) and by the yield strength of the tissues involved. We use 'stall' here in the sense that cellular convergence can no longer proceed, e.g. because the tension across individual cells engaged in MIB is high enough that they can no longer pull themselves between each other; it is less clear what factor limits convergence by CT. Once the yield stress (force/area) for a tissue is reached, forces generated by CT and CE result in plastic deformation of that tissue, and convergence in one region is balanced by strain in another. Understanding the behavior of the explant requires an understanding of the biomechanically complex structure of the explant, and its dynamic changes over time. Also, it may be assumed that yield strength and stall force as they apply to the explant are not 'all or nothing' effects; there are likely to be multiple structures with different rates of viscous flow leading to different rates of plastic deformation across a range of tensions. And because each cell is a motor, and is differently arranged within the tissue, they will reach their stall force at different over-all tensions across the explant.

We propose that initially (first 20–30 min) convergence proceeds rapidly until the stall force of the machine(s) is reached. At this point, force increase slows, advancing only as additional cells are recruited, either by the progression of MIB into more posterior tissues, or as shear allows convergence to proceed such that more MIB expressing cells are acting in parallel (see *Figure 1I,J*), increasing the 'instantaneous' stall force for the current extent of morphogenesis. It less clear what effect convergence should be expected to have on an increase of the overall stall force of CT, but our results from ventralized explants (*Figure 3D*) suggest that they too increase stall force with convergence. This slower rate of force advance continues until the plateau, at which point the level of tension across the explant reaches the yield stress for the LV region, which results in its observed slight increase in strain rate, while CE continues to drive convergence in more dorsal tissues.

Our model above, that encumbered explants are increasing tensile force as they recruit more cells into MIB, predicts that applying additional exogenous tension to an explant should prevent further force generation by the explant until shear has allowed enough convergence such that the instantaneous stall force rises above the current level of tension. To test this, we applied additional tension to explants at various stages through the end of the plateau, by increasing the strain on them (*Figure 3—figure supplement 4A–E*). Explants (n = 9) generally showed an immediate 0.5 to 0.6 µN increase in tension, which decayed quickly over the next 15 to 30 min, then remained static until the explant's projected rise in force prior to being strained reached its new, current level of tension (e.g. *Figure 3—figure supplement 4A,B*). In no case did tensioned explants produce higher final amounts, or rates of increase, of force. These results demonstrate that increased tension stalls the force increase, as predicted, and that while the explant can sustain greater tension, increased tension alone does not trigger increased force production.

If the force developed by the explant at a given time does represent an instantaneous stall force, we expect that decreasing tension by decreasing strain should allow more rapid convergence until the system maximum is reached again. Reducing strain by moving the anchor toward the probe (*Figure 3—figure supplement 4F–I*) enough to decrease tension on the explant by 1.2 to 1.4 µN resulted in an immediate observed relaxation across the explant of from 0.6 to 1.2 µN, with the remainder corresponding to the rapid recoil of elastic strain during anchor movement (<1 s). After the initial, rapid elastic recoil, explants converged at a rate similar to unencumbered explants (about 10 %/hr, see *Table 1*), until recovering their prior tension levels (e.g. *Figure 3—figure supplement 4F,G*). Recovery was consistently (n = 6) rapid (<15 min) during the first phase of force increase, whereas it was consistently (n = 4) slower (30 min or more) during the major plateau or during dorsal bottle cell re-expansion. After recovery during the first phase of force increase, explants converged at rates more typical of encumbered explants (about 4 %/hr, *Table 1*). This rapid recovery supports our model that explants generate increasing tension until reaching an instantaneous stall force, with

consequently retarded convergence during all but their initial period of force increase. The slower recovery of the explants to normal levels of tension during the plateau is consistent with lower levels of stored elastic energy, across the explant as a whole at this time (Discussion).

We tested the idea that less force was being generated during the plateau phase, based on the premise that reduced force generation results in lower stored elastic energy, which is presumably continuously dissipated by long term viscous tissue flow, loss to heat, etc. While not quantitative, the immediate, rapid elastic recoil (tension released by anchor movement – observed tension drop; see *Figure 3—figure supplement 4F–I*), which occurs at a rate much higher than observed for normal explant convergence, suggesting it is not dependent on metabolic energy expenditure (*Chen, 1981*), should provide a qualitative assessment of the relative amount of elastic energy stored. Both the fraction of tension released by anchor movement that was recovered during rapid recoil and the rate of that recoil were greater during the first phase of force increase compared to the plateau phase. We compared the fraction of tension relaxation that was recovered within the first 5 s (fractional recoil = immediate elastic recoil/tension released by anchor movement; see *Figure 3—figure supplement 4F–I*), and the rate of recoil in the first 5 s (recoil rate = immediate recoil distance (as a percent of total mediolateral explant width)/time). We found that the fractional recoil was 41% (S.E. =±5%, n = 6) during the first phase of force production, compared to 28% (S.E. = ±5%, n = 3) during the plateau. The recoil rate was 670 %/hr (S.E. = ±90 %/hr; n = 7) during the first phase of force increase (similar to the recoil rate of 600 %/hr seen in explants freshly cut from intact early gastrulae), compared to 390 %/hr (S.E. = ±32 %/hr; n = 3) during the plateau. These results show that during a tractor pull, explants, like the intact embryo, store considerable elastic strain-energy, more of which is recoverable during the first phase of force increase than during the subsequent plateau. This is consistent with the idea that ventrolateral tissues have reduced force accumulation during the plateau (Discussion). Alternatively, the rate of dissipation may increase during the plateau, e.g. because the tissue has reached its yield stress and is deforming plastically.

## Structural stiffness increases in all tissues around the end of gastrulation

In order to develop better models for how embryonic tissues deform in response to the forces they generate, it is useful to measure embryonic tissue stiffness. As a general concept, stiffness is measured by asking how much a material stretches along the axis of tension (strains) when a known amount of tensile stress (force per unit area) is applied to it. Structural stiffness and the elastic modulus are different specific mechanical properties that describe a material's stiffness and that can be measured experimentally with similar methods. The **elastic modulus** is a fundamental property of a particular type material, for example steel, independent of its specific geometry, and is measured by asking how much a uniform material of known cross-sectional area (orthogonal to the axis of tension) strains when a known amount of tensile stress is applied to it. By contrast, **structural stiffness** of a composite structure depends on the mechanical properties and arrangement of the individual elements that make up the structure, such as a bridge made of steel beams, or in our case, of the individual tissues and cell types and ECM that make up the explant. Elastic modulus of the individual elements may be inferred from a measurement of structural stiffness if the elements and architecture of the structure can be accurately modeled and the strains in each element measured, an approach that has been applied to embryonic tissues (e.g. *Zhou et al., 2009*), but is beyond the scope of the current work.

We could not measure a true elastic modulus of embryonic tissues because the tests are not isometric nor are the explants uniform or homogenous (see Appendix 5). Instead we estimated structural stiffness along the mediolateral axis of the explants with a uniaxial, tensile stress relaxation test (*Figure 4*), and estimated sagittal sectional areas from fixed samples (*Figure 5A*; see Appendix 3). In our case, structural stiffness reflects the mechanical properties and organization of the individual tissues and cell types that make up the explant, as well as its overall geometry. Fixed samples of standard Giant and Dorsal 180° explants show similar increases in sagittal area (*Figure 5A*) due to progressive MIB-mediated intercalation (*Figure 1I,J*). Ventral 180° explants increase in sagittal area until about the time of the plateau, then remain stable, suggesting either that thickening has reached an equilibrium, or is developmentally programmed to stop.

From the time-dependent stress decay (*Figure 4C,F*), parameters for a network model of stress relaxation were estimated (*Figure 4E*; see Appendix 4), including the residual structural stiffness and

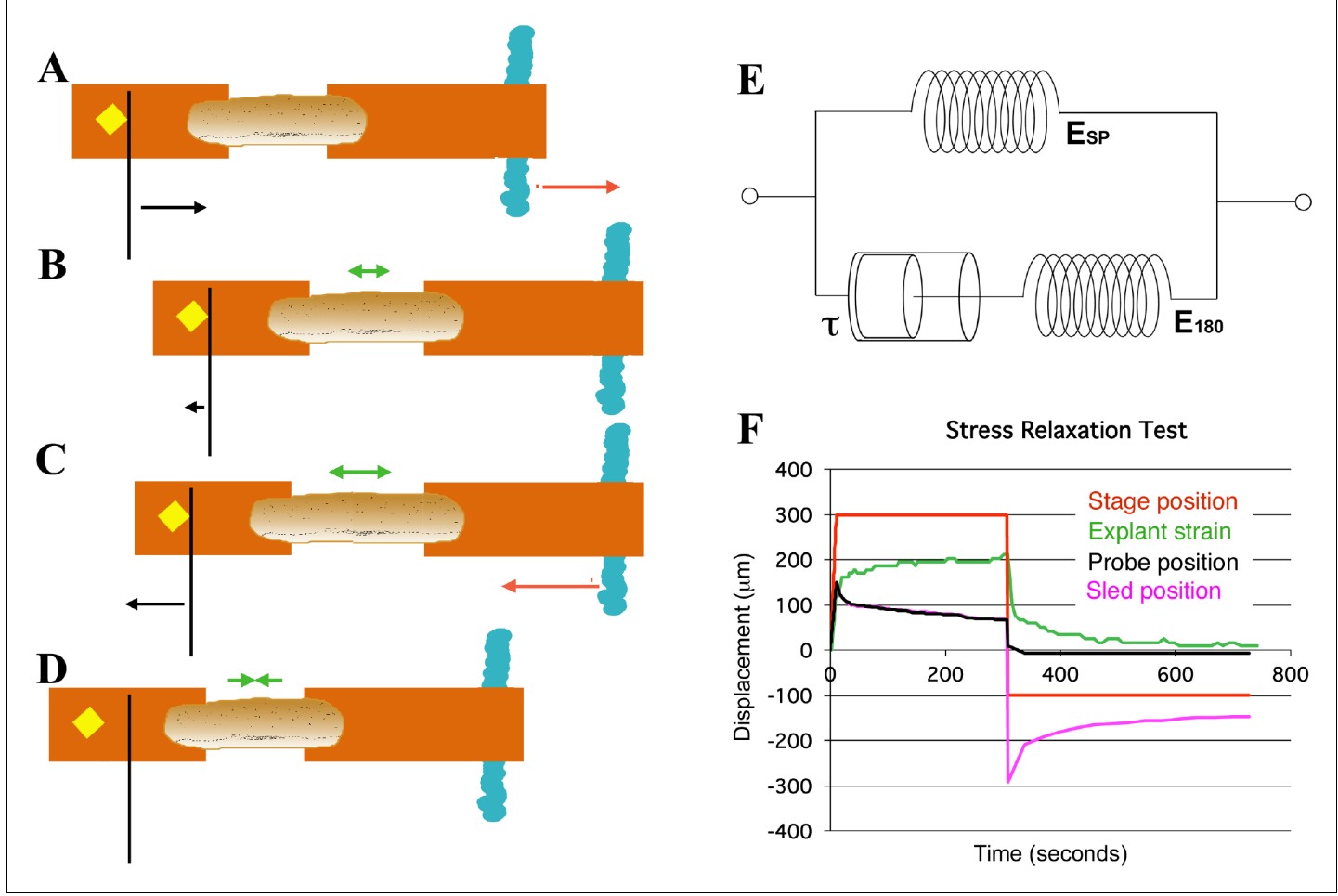

**Figure 4.** Schema of movements and measures involved in stress-relaxation test. Start position, with cleat adjacent to probe. Stage is moved 300 microns (**A**, red arrow) against probe, to impose stress, with resulting probe deflection (**A**, black arrow). The explant shows an instantaneous strain (**B**, green arrow), then exhibits viscoelastic decay, or 'relaxation' over time (**C**, green arrows), reducing the deflection of the probe (**B**, black arrow), until tension equals residual stiffness (in practice, $E_{180}$). Finally, the stage is moved back 400 microns (**C**, red arrow), which de-stresses the explant and allows the probe to return to its starting position (**C**, black arrow). The explant shows elastic recovery (**D**, green arrows). (**E**) A model of the explant as a viscoelastic material, with springs representing instantaneous ($E_{SP}$) and residual ($E_{180}$) stiffness, and a dashpot representing the viscosity, with relaxation time (half-time of decay), tau ($\tau$). In an example of a stress-relaxation test (**F**), the stage, to which the anchor strip (left) is attached, is moved (**F**, red line) to impose a stress, by pulling the cleat against the probe (as in **A**, **B**). This imposes a strain (**F**, green line) on the explant, and deflects the probe (**F**, black line), as in (**A**). The explant continues to undergo strain, as in **C**, until it reaches its residual stiffness. The movement of the sled (**F**, magenta) initially parallels that of the probe, until the stage is moved away from the probe (as in **C**, **D**) at about 300 s, at which point the explant shows elastic recovery of the imposed strain (as in **D**), pulling the sled with it. In order to estimate $E_{sp}$ and $\tau$ we used non-linear regression curve fitting of the stress relaxation phase (**B**, **C**, green arrows; **F**, green line).

DOI: https://doi.org/10.7554/eLife.26944.024

The following source data is available for figure 4:

**Source data 1.** Source data for Stress Test Example, Panel F.
DOI: https://doi.org/10.7554/eLife.26944.025

constant of spring stiffness for the explant at 180 s after strain application (*Figure 5*), as well as instantaneous structural stiffness and viscosity (*Figure 5—figure supplement 1*; Appendix 4). A standard 300 μm displacement from the probe produced an average 12% (range = 8% to 14%) strain of the mesodermal region between the sleds by 180 s, in giant explants initially at rest. The modulus estimated from structural stiffness's along the mediolateral axis of giant explants rose significantly from 14 Pa (Pascals) at late gastrulation (G+4.8hr) to 21 Pa by mid neurulation (G+8.7hr) (p<0.01, paired t-test, n = 6 vs. 6) (*Figure 5B*). After release from tension, explants recoil rapidly and typically

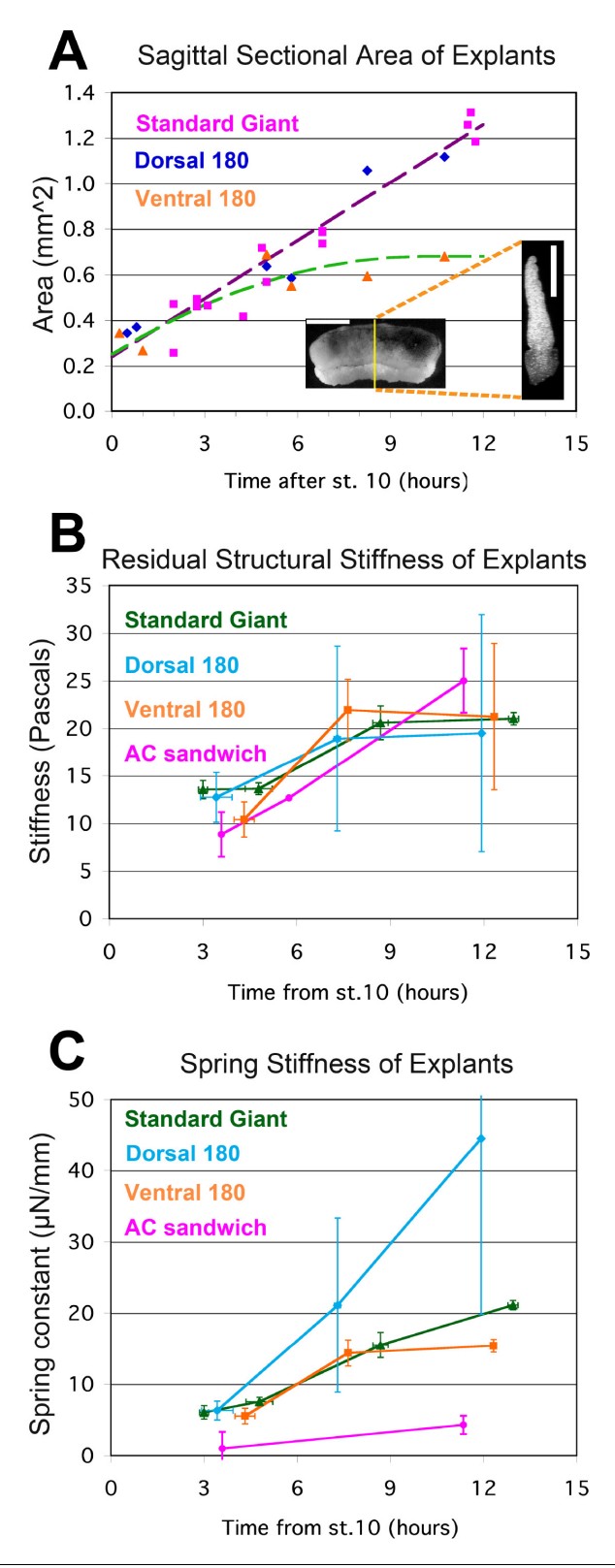

**Figure 5.** Stress-relaxation tests. The sagittal sectional area (SSA) for different kinds of unencumbered explants was determined at points throughout gastrulation and neurulation from confocal z-series of RDA labeled explants (A, inset; scale bar = 1 mm in intact giant, 0.5 mm for sagittal cross section (at yellow line in giant)). Standard giant (magenta squares) and Dorsal 180° sandwich explants (blue diamonds) show similar progressions of SSA; a

*Figure 5 continued on next page*

*Figure 5 continued*

regression on both (violet dashed regression line, 0.085 mm²/hour * (hours after G0)+0.24 mm²) was used to estimate stage specific SSA for both kinds of explants in stress-relaxation tests. The increase in SSA for V180° explants tended to plateau by about 8 to 10 hr, so a first order polynomial regression (green dashed line, −0.0042 mm²/hour² * (hours after G0)² + 0.085 mm²/hour * (hours after G0) +0.25 mm², or 0.68 mm² at 10 hr or later) was use to estimate the stage specific SSA for V180° explants in stress-relaxation tests. The SSA of animal cap explant sandwiches was a consistent 0.16 mm². Estimated SSA, measured force on the probe at 180 s, and measured proportional strain on the explant at 180 s were used to determine the stiffness ($E_{180}$) at several times during gastrulation and neurulation (**B**). Standard giant sandwich explants (dark green line), as well as Dorsal 180° (light blue) and Ventral 180° (orange) sandwich explants and animal cap sandwich explants (Yellow) were tested. In order to compare the force-bearing capacity of different tissues, a bulk spring stiffness (Force at 180 s / Strain at 180 s) was plotted (**C**). Error bars = standard error of the mean, n's = 3 to 6, except where no error bar is shown, where n = 1.

DOI: https://doi.org/10.7554/eLife.26944.026

The following source data and figure supplements are available for figure 5:

**Source data 1.** Source data for Sagittal Sectional Area (**A**) and Structural Stiffness and Spring Stiffness (**B**, **C**), on separate panels within Excel file.
DOI: https://doi.org/10.7554/eLife.26944.029

**Figure supplement 1.** Instantaneous Structural Stiffness and Viscosity of standard giant sandwich explants.
DOI: https://doi.org/10.7554/eLife.26944.027

**Figure supplement 1—source data 1.** Source data for Instantaneous Stiffness and Viscosity measures.
DOI: https://doi.org/10.7554/eLife.26944.028

returned to near their initial width within three minutes, with about 25% of the initially applied strain lost to permanent, or plastic, deformation, leaving them about 3% longer, indicating that while they can store elastic strain energy, some dissipation occurs, for example by the rearrangement of cells or structural elements within them.

D180° and V180° sandwich explants also showed a trend of increasing stiffness from gastrula to neurula stages (*Figure 5B*) with both showing an increase between stages 12 and 14 (G+4.3 and 7.6hr) but showed no significant differences from each other at any stage. Presumptive ectodermal (AC) sandwiches strained an average of 24% over 180 s showed increased structural stiffness (p<0.01, n = 5 vs. 3) from gastrula to neurula stages (*Figure 5B*), but their stiffness is not significantly different from that of standard giants at either stage. By late neurulation AC explants also showed plastic deformations of 25% of total strain. AC explants were substantially more plastic during gastrulation however, with about 65% of the total strain remaining as plastic deformation, suggesting a lesser ability to store elastic strain energy than during neurulation.

Although a bulk measure for the explant as a whole, our measures of structural stiffness provide baseline estimates for mechanical behaviors of embryonic tissues. Somewhat surprisingly, all explants types measured, although comprised of different sets of tissues, exhibit similar structural stiffness, and their stiffness increases by about the same amount from gastrulation to neurulation.

To understand how differential strain of tissues, as found above in our investigation of the plateau, relates to blastopore closure, we measured spring stiffness for each explant type over time. Tissue spring stiffness (force/strain) reflects the deformation or strain that a tissue of a specific cross sectional area will undergo in response to a force, and so can be used to predict the strain in different tissues across the explant when subjected to the same tension. Standard giants, D180°, V180° and AC explants all showed a trend of increasing spring stiffness between gastrula and neurula stages (*Figure 5C*). Standard giants did not differ significantly between gastrula stages but increased significantly from stage to stage thereafter (p<0.01, n's = 6 vs. 6; 6 vs. 3), doubling between G+4.8 and G+8.7. AC sandwich explants also increased significantly (p<0.05, n = 3 vs. 3) between gastrula and neurula stages. Standard giants had significantly higher spring stiffness than AC sandwich explants by 5 to 6 fold at all stages (p<0.05 during gastrulation, p<0.01 during neurulation; n's = 3 vs. 3 to 6), indicating that ectodermal tissues contribute little to the ability of the IMZ to resist tension along the mediolateral axis during tractor pulls. D180° explants showed greater increases compared to V180° explants and standard giants, but were not significantly different at any stage. V180° explants were similar to standard giants until after mid-neurulation when their spring stiffness was

moderately (37%) but significantly lower (p<0.01, n = 2 vs. 3). The greater spring stiffness of giants vs. ventral 180° explants by the end of neurulation is consistent with increased overall spring stiffness of giants as dorsal tissue differentiation progresses laterally. In the context of our sandwich explants by the time of the plateau, ventrolateral tissues undergo more strain than dorsal tissues, because dorsal tissues have a larger and increasing sectional area.

## Discussion

### The circumblastoporal tissues (IMZ and NIMZ) produce and maintain long-term, long range, consistent patterns of convergence force throughout early development

Giant sandwich explants (IMZ and NIMZ) generate and maintain a consistent pattern of convergence force throughout gastrulation, neurulation, and into the tailbud stage. Correction for drift of the two probes of different stiffness's yielded similar force profiles, suggesting that our results are robust to different sources of error. Wound healing at the edges of the explants (*Davidson et al., 2002*) or a response to cell lysate from surgery (*Joshi et al., 2010*; *Kim et al., 2014*) could generate force, but AC explants, which are also cut and heal, do not generate significant tensile force, making this unlikely.

### CT generates preinvolution, circumblastoporal tension throughout gastrulation

Previously, post-involution CE was thought to generate the convergence force driving blastopore closure, as well as the post-involution anterior-posterior extension that elongates the body axis (*Keller and Danilchik, 1988*; *Moore et al., 1995b*; *Keller et al., 2000*). CT was described in the ventral sector of the *Xenopus* gastrula (*Keller and Danilchik, 1988*) but its force contribution to gastrulation and the fact that it occurred everywhere in the IMZ was unknown. Here, several results show that CT generates circumblastoporal tension early and throughout the IMZ. First, standard giant explants produce force early (*Figure 3B*), before MIB and CE have begun (G+2hr, stage10.5) (*Shih and Keller, 1992b*; *Lane and Keller, 1997*). Second, ventral 180° and ventralized giant explants that do not express CE also generate circumblastoporal force (*Figure 3D*), which is likely due to their expression of CT. Third, unencumbered ventralized giants show uniform CT throughout the MZ (*Video 3*). Thus CT generates all the IMZ convergence prior to the onset of CE at the early midgastrula stage (stage 10.5, G+2hr). Although force from ventral and ventralized explants levels off during the plateau period of standard giants, it persists and contributes to blastopore closure throughout gastrulation by decreasing the circumference of the IMZ and directing it to the point of involution. CT may continue into neurulation and function in the late involution of the ventrally located, posterior paraxial mesoderm (*Keller and Tibbetts, 1989*; *Wilson et al., 1989*). These findings explain how ventralized *Xenopus* embryos (*Scharf and Gerhart, 1980*) and normal embryos of some amphibians such as *Gastrotheca riobambae* (*del Pino, 1996*) close their blastopores in the absence of CE (*Keller and Shook, 2004*; *del Pino et al., 2007*) and symmetrically, as are CT movements in normal *Xenopus* embryos. These results establish CT as a morphogenic machine independent of CE, and raise the question of how widely it occurs and how it is integrated with other movements, in amphibians as well as other species, (see below).

### The force profile reflects the transition from CT to CE and illuminates the mechanics of gastrulation

Analysis of the force profiles of the various explants, in the context of the degree to which they express CT and CE, illuminates much about the mechanics of gastrulation and blastopore closure. We represent the capacity of CT and CE to generate and transmit tensile force in the explants at different stages of expression of CT and CE (A-D) as a linear array of motors and springs (light ones representing CT, darker ones CE/MIB, *Figure 6B'–D'*)). First, the fact that these tissues can generate, transmit, and maintain tensile force across the millimeters of tissue (*Figure 6E*) indicates that there is a global, large-scale mechanical integration of forces, meaning that morphogenic movements anywhere throughout the 1.2 mm diameter embryo could affect one another mechanically. The tension generated by the motor and the stiffness of the spring determines the convergence or

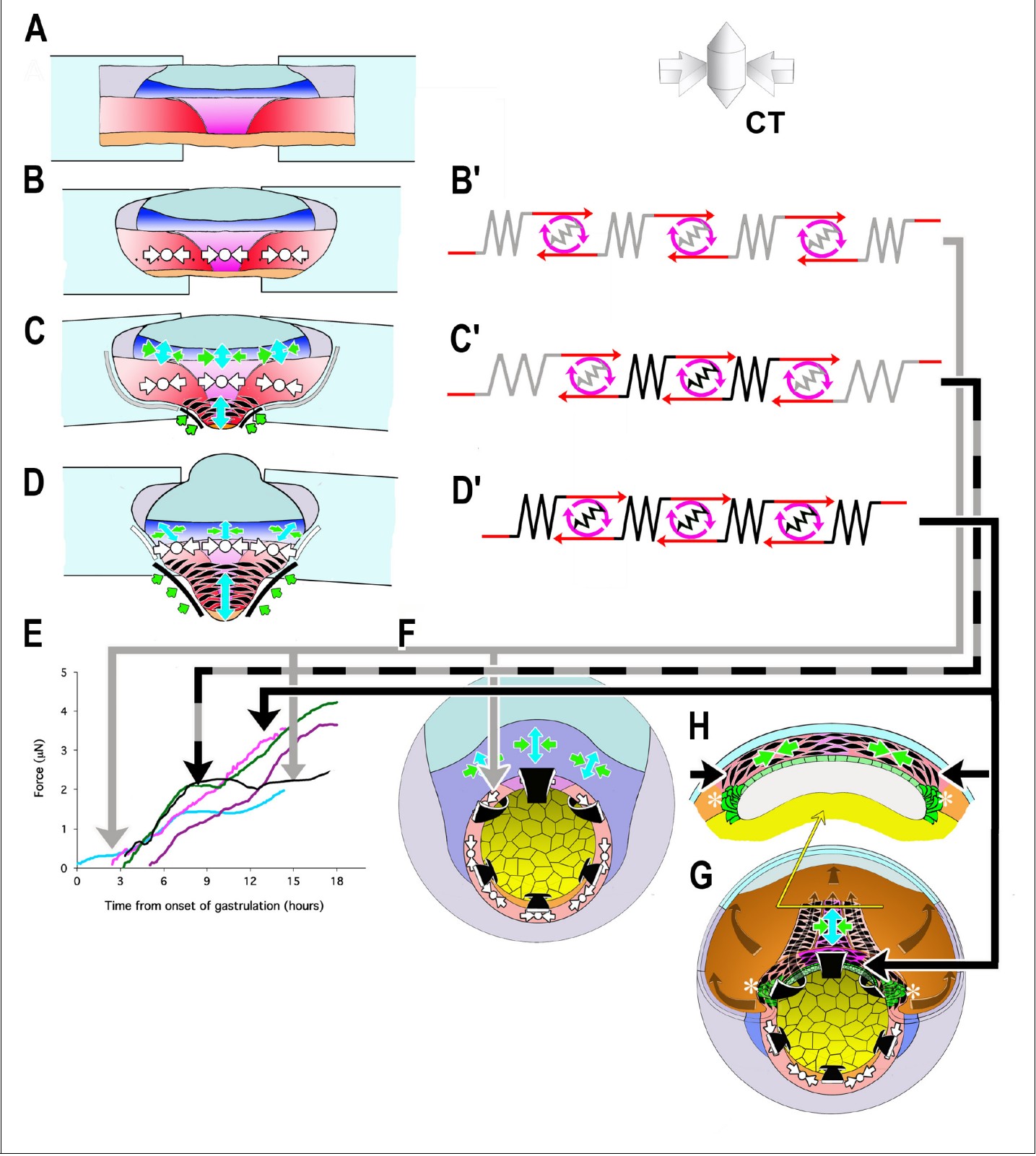

**Figure 6.** Model of how the convergence forces measured in the tractor pull are related to modules of cell behavior in explants and embryos. Early convergence forces are generated largely by the CT machine (CT symbols, (B–D). The CT symbol implies ML tensile force, represented by the inward pointing arrows, and radial compressive force, represented by the dot and indicating force directed in and out along the radial axis of the embryo (see 3D representation of CT). As CE begins, MIB (fusiform, black cells) progressively replaces the CT machine from presumptive anterior to posterior (CE

*Figure 6 continued on next page*

*Figure 6 continued*

symbol: green convergence, blue extension arrows, (**C, D**) while CT continues in more posterior tissues (CT symbol, **C, D**). At or shortly after the onset of mesodermal MIB, MIB and CE begin in the posterior neural tissue (blue tissue, CE symbols, (**C, D**). Thus the IMZ tissues express a changing combination of CT and CE as development progresses. We represent CT and CE as modules, expressing distinctive spring constants (grey or black coils) and motor strengths (red and grey or black symbols), with the lighter spring and motor indicating CT and the darker CE (**B'–D'**). Force plots in (**E**) are means from *Figure 3*: blue = Early Giants; pink = Dorsal 180 s; green = Standard Giants, Probe #4; purple = Late Giants; black = ventralized giants. Initially, up to about G+2hr, the entire IMZ is comprised of CT-modules (**B'**), which represents the situation and generates the forces seen in the first two hours of early pulls (follow grey arrow from B' to E, blue line). These forces likely approximate force generated in the preinvolution (uninvoluted) IMZ of the whole embryo (follow the gray arrow to F, CT symbols). As MIB begins, a CE module lies centrally, flanked by CT-modules in series (**C'**) with lateral edges attached to the strips (**C**), which represents the situation and generates the forces measured from G+2 to 10.5hr, including the period during the plateau in standard pulls (follow black/gray dashed arrows to E, green line), but with an increasing contribution from CE vs. CT modules after G+2hr. As more cells express MIB, the number pulling in parallel increases, increasing the spring constant and motor strength of the CE module. Because the VAZ forms as an arc it does not initially pull directly on the attachment strips (**C**; green arrows at edge) but on the intervening CT modules. The nascent CE module is initially small and weaker than the adjacent CT modules, but becomes larger over time, resulting in both increasing spring constant and motor strength. This eventually overpowers the shrinking CT modules, such that their convergence is reduced (**C'**, more open coils), which dissipates some of the tension generated by the CE module, and thereby contributing to the plateau. In contrast, in the embryo, the CE (MIB) module is, from the beginning, always anchored to the endoderm at both ends, with only an indirect connection to CT modules in the lateral and ventral portion of the MZ (not shown in F; see H, G, asterisks). Thus CT acts as a continuous but diminishing ring of converging tissue outside the blastopore, while CE-expressing tissue inside the blastopore, primarily in series with the relatively inert endodermal tissue, acts in parallel with this ring. The transient decline rather than plateau in the rate of force increase during late pulls (E, purple line) can be explained by a larger domain comprised of CE modules and smaller domains comprised of CT modules, compared to standard control pulls at the onset of the plateau, such that not all force generated by the CE domain was absorbed by reduced convergence in the CT domain. Once MIB progresses laterally to points of attachment with the strips (**D, D'**), the decline ends; this represents the situation during the second phase of force increase (follow black arrow to E, magenta line and to H, (**G**). At this point, all the force generated by CE and MIB in posterior tissues are transmitted to the attachment strips, while, with the progression of MIB posteriorly, force from more anterior tissues is transmitted progressively more indirectly, at an angle (green arrows; **D**). Dorsal pulls show no plateau, because they contain little or no tissue comprised of CT modules (**D'**) by the onset of the plateau (follow solid black arrow to E, pink line). Ventralized giant pulls show maximal force generation at the level of the plateau, being comprised of only CT modules (**B'**; follow the gray line to E, black line).

DOI: https://doi.org/10.7554/eLife.26944.030

strain experienced by each element of the array as well as neighboring elements and, integrated across the explant, of the explant as a whole. Second, based on the mediolaterally uniform convergence during CT, prior to the onset of CE (*Shook et al., 2018*), we assume that mechanical properties and behavior within the IMZ are uniform before CE begins, and the evidence shows that CT is expressed uniformly across the mediolateral extent of the explant, prior to the onset of CE and thus accounts for all the measured force up to G+2hr (*Figure 6B'*, gray arrow to E, blue force trace of early giants). The same is true of ventralized embryos, expressing CT throughout the IMZ (*Figure 6B'*, gray arrow to E, black force trace of ventralized embryos). The timing of CT in explants argues that this force occurs in the pre-involution IMZ of the embryo (*Figure 6B*, gray arrow to 6F, embryonic preinvolution IMZ), and its uniform dorsoventral expression accounts for the relatively uniform convergence of the blastopore during early gastrulation of normal embryos and throughout gastrulation in ventralized embryos (*Figure 1A,N*, green arrows).

At stage 10.5, cells begin to undergo a progressive, patterned transition from CT to MIB (CE) as they involute, beginning in the presumptive anterior somitic mesoderm with formation of the Vegetal Alignment Zone (VAZ), and progressing medially and posteriorly (*Shih and Keller, 1992b*) (*Figure 6C*). At this point the explant becomes morphogenically heterogeneous, with a presumptive anterior region of MIB driving CE located centrally (*Figure 6C*, black fusiform cells, green, blue arrows), and a presumptive posterior region of CT uniformly expressed across the remaining posterior of the IMZ (*Figure 6C*, white symbols). In this phase, the central CE/MIB region is initially linked to the sled and anchor strips only via the lateral, CT expressing regions of the explant (*Figure 6C, C'*), and force generation by the central region of MIB results in the observed, continued convergence of this region while the lateral region expressing CT, with its lower spring stiffness, undergoes strain. The weaker spring-motor combination of the CT expressing regions thus limits the transmission of force to the measuring apparatus (*Figure 6C'*, gray +black arrow to E, plateau in green force trace), eventually leading to the observed plateau (*Figure 3*), when the lower yield strength of the CT region (as reflected in its lower spring stiffness) is reached. As the transition of MIB/CE spreads further posteriorly, it reaches the sled and anchor strips at the end of the plateau (G+10.5hr), the

weaker CT regions are replaced by stronger MIB/CE regions (as reflected by their higher spring stiffness), the weaker link is removed, and the force rises again (*Figure 6C–D,C'–D'*, black arrow to E, green line in E). This interpretation is supported by our observations that ventral giant explants, which only express CT, reach but never advance beyond the same plateau level of force as standard giant explants containing the lateral regions (*Figure 6E*, black and green force traces, respectively), that dorsal explants, which have MIB directly connected to the sleds, show no plateau and that late giants, which already express MIB across a greater mediolateral span of the marginal zone when initially placed under tension, show only moderately decreased force increase during the plateau period of standard explants (*Figure 6E*, pink and purple force traces, respectively).

In the embryo, however, the stronger MIB/CE region never acts in direct series with an intervening, weaker CT region; instead MIB originates at the lateral boundary of the somitic mesoderm at its junction with the vegetal endoderm, and is anchored there (*Figure 6G,H*, asterisks), thereby forming a continuous, uniform convergence mechanism based on CE (D'), which acts as a coherent system in the embryo (*Figure 6D'*, black arrow to G, H), a situation achieved in the post-plateau explant (*Figure 6D*, black line to 6E, post-plateau standard giant, green, dorsal 180 explants (pink), and late explants (magenta) force traces).

The influence of the post-involution expansion of MIB expression posteriorly from its onset in the VAZ is reflected in the progressively more anisotropic blastopore closure from stage 10.5 onward, and in the dorsal region of the IMZ dominating closure, something that cannot be accounted for by the isotropic convergence of pre-involution CT (*Figure 1A,D*, green arrows).

Unlike CT, the total force generated by CE (MIB) is under-estimated by our measurements, because CE results in extension of many of the cells expressing MIB away from the zone directly between the sleds, and these cells therefore pull on the sled/anchor system at an increasing angle (*Figure 6D*). However, the same is true in the embryo, as progressively more of the population of MIB expressing cells lie far anterior of the posterior progress zone of MIB at blastoporal lip (*Keller, 1984*; *Keller et al., 1989*; *Keller and Tibbetts, 1989*; *Wilson and Keller, 1991*)(*Figure 6G,H*), and thus the force measured with the giant explant may reasonably approximate that generated at the blastoporal lip through the end of gastrulation.

The convergence of the somitic mesoderm during late neurulation involves MIB but also columnarization (thickening) (see *Keller et al., 1989*; *Wilson et al., 1989*), which forms converging 'somitic buttresses' that may contribute to folding neural plate (*Schroeder, 1971*; *Keller, 1976*). Our force measurements of convergence forces exerted by the intact embryo during this time are unquestionably a substantial underestimate (see also Estimates of Force/cell, below), both for the reasons listed above, and because embryos have assembled their mesoderm and neural tissues into laminar aligned structures that have undergone the full extent of normal convergence movements, in contrast to the retarded convergence of our explants.

Finally, when we explant tissue it expends its stored elastic energy as it converges rapidly, and consequently its subsequent convergence is slower than intact embryos and the additional force is measured. Observed force is further reduced by friction, perhaps by 0.2 µN, despite the slick agarose pad beneath the explant and by the glass beads beneath the sled. Therefore force measured here should moderately underestimate that generated at the instantaneous stall force or yield strength of the embryo, for a given extent of morphogenic progress.

## Explant stiffness

*Xenopus* embryonic tissues stiffen about two fold along their mediolateral axis around the end of gastrulation, regardless of the tissue type, suggesting a systemic mechanism. Such increases may have many causes, such as increased ECM deposition (*Davidson et al., 2004*; *Skoglund et al., 2006*), increased cell-cell adhesion, or stiffness of cytoskeletal architecture (*Zhou et al., 2009*). The high plasticity after strain of the ectodermal tissue during gastrulation is consistent with its response to strain during epiboly, when its area is increased by about two fold (*Keller, 1975*; *Chien et al., 2015*). Whatever the causes, the increase in the stiffness of all the tissues at the end of gastrulation may cause decreased plasticity, as well as the resetting of pseudo-elasticity to a thinner epithelial set point (*Luu et al., 2011*). Previous estimates of AP compressive stiffness of dorsal isolates at stage 11.5 (about 14 Pascals, *Moore et al., 1995a*; *Zhou et al., 2009*), are very similar to ML tensile structural stiffness at stage 11.5 (G+3.5hr) measured here (*Figure 5B*), suggesting that the same mechanical elements may be resisting ML widening in both cases. In explants, the constant of spring

stiffness for dorsal tissue continues to rise after early neurulation while that for ventral tissue does not (*Figure 5C*), in part because the transverse sectional area of tissue between the attachment strips increases in the former but not in the later (*Figure 5A*). More reliable measurements will be required to properly resolve the relative stiffness of ventrolateral or posterior tissues expressing CT compared to dorsal tissues expressing CE. In any case, the observed transmission of forces across the full length of the IMZ makes it clear that tissue stiffness links the entire embryo into a mechanical 'mechanome' (*Kamm, 2006*; *Lang, 2007*) in which major regional morphogenic movements can affect one another. This accounts for the fact that molecular interdictions of epiboly of the animal cap can affect blastopore closure on the other pole of the embryo, although no molecular perturbation was made there, and that the blastopore closure defect can be rescued by cutting off the animal cap, thus breaking the mechanical link (e.g. *Eagleson et al., 2015*). Lastly, the tissue connectivity of force-producing regional morphogenic movements is an important parameter for successful embryonic morphogenesis. For example, direct linkage of MIB/CE to the sleds, rather than linkage through CT regions, eliminates the plateau in measured force increase, suggesting that the direct anchor-points of MIB in the vegetal endoderm in the embryo are an important aspect of its architecture.

## Accommodation to load and stall force

Some evidence suggests that tissues modulate their force production in response to changes in load. The mechanical properties of embryonic tissues from different clutches vary (*von Dassow and Davidson, 2009*), yet gastrulation proceeds at roughly the same rate, suggesting that force production accommodates to the tissue properties encountered. Explants of dorsal tissues embedded in gels of increasing stiffness respond by producing more force (*Zhou et al., 2015*). Time-lapse recordings of blastopore closure show occasional decreases in rate, including temporarily stalling out and then recovering rapidly, as if transient overloads of resistance were being overcome by increased force production (personal observation). However, in our experiments increasing tension did not result in greater convergence force, and instead, temporarily stalled force increase until further morphogenesis had occurred. We suggest that this is because our explants are already at their instantaneous stall force (see below and Appendix 6).

Our results suggest that explanted tissue builds tension relatively rapidly when initially encumbered, until it reaches its stall force. Further force increase is then limited by the rate at which the number of cells expressing MIB increases and by their rate of intercalation, such that more pull in parallel, rather than in series. Intercalation is in turn limited by the roughly 4 %/hr rate of convergence allowed by explant shear off the attachment strips, lateral strain, and, to a much smaller extent, probe movement. We predict that allowing more rapid convergence should allow force to rise more rapidly. It is not clear how force generation by CT would be expected to change as tissue thickens, but the ventral/ventralized tractor pulls show that force increase correlates with thickening, until both cease once reaching the plateau.

Mesodermal tissues in normal embryos are probably rarely at their stall force, since convergence occurs more rapidly. When convergence is impeded, tension comparable to that generated in tractor pulls accumulates around the blastopore during gastrulation (*Feroze et al., 2015*), suggesting these are also measures of force at their instantaneous stall force. A more accurate reflection of forces in the embryo might be obtained by looking at points along the force-velocity curve more closely resembling the situation in the embryo, e.g. by starting with about 0.2 μN of tension and moving the anchor strip toward the probe at 1 or 2 %/hr.

## Estimation of forces generated per cell and tensional stress of convergence

To estimate the force generated per engaged cell, we determined the average effective sagittal-sectional area (SSA) of the deep mesoderm, the cell population we expect is effective in bulk force production during the tractor pull (*Table 2*). We estimate that the mean mediolateral tensile force per cell rises during the first half of gastrulation and stabilizes by mid gastrulation at around 2.3 nN/cell, where it remains (*Table 2*). These values assume negligible contribution from neural tissues and give equal weight to each cell within the effective SSA, although different proportions of cells may express CT, CE or no force generating behavior and may direct force mediolaterally more or less efficiently. We assume a constant cell size, although some cell division occurs in the somitic

**Table 2.** Estimates of force per cell and tensional stress within effective sagittal section area (SSA) (deep mesoderm only). Based on a mean cell sectional area of 625 nm$^2$.

| Time from onset of gastrulation (hours) | Force (μN (n,±SEM)) | Effective SSA (mm$^2$ (n,±SEM)) | Estimated force/cell (nN) | Force per effective SSA (Pascals) |
|---|---|---|---|---|
| 1 | 0.25 (3, 0.08) | 0.12 (1, n/a) | 1.3 | 2.1 |
| 2.1 | 0.31 (3, 0.09) | 0.11 (4, 0.014) | 1.7 | 2.8 |
| 2.9 | 0.49 (4, 0.16) | 0.13 (4, 0.018) | 2.3 | 3.8 |
| 4.3 | 0.94 (4, 0.04) | 0.27 (3, 0.024) | 2.2 | 3.5 |
| 6.5 | 1.6 (5, 0.11) | 0.41 (3, 0.020) | 2.5 | 3.9 |
| 11.8 | 2.6 (6, 0.19) | 0.68 (3, 0.051) | 2.4 | 3.8 |

DOI: https://doi.org/10.7554/eLife.26944.031

mesoderm during gastrulation and neurulation. The constant force per cell is consistent with the idea that increasing stage-specific maximal force generation is limited primarily by morphogenesis, as it increases the SSA. We also calculated tensional stress within the effective SSA (*Table 2*), which was roughly 4-fold lower than the mediolateral tensile stress estimated from extension forces exerted by dorsal tissues in a gel (*Zhou et al., 2015*), which is not surprising given that their measurements capture all of the force generated during late neurula stages in tissue that has undergone normal morphogenesis, compared to our which primarily capture forces generated around the blastopore, as described above.

## CT to CE, a major morphogenic, regulatory, and evolutionary transition in the Amphibia

The temporal, spatial parameters of the CT to CE transition, and its biomechanical implications, represent a major morphogenic and regulatory transition. The force traces show that CT-generated forces occur early and throughout the IMZ prior to its involution, whereas CE, and its underlying cell behavior, MIB, are expressed after involution and progressively, with increasing numbers of cells acting in parallel with time (*Shih and Keller, 1992b*). This progressive CT to CE transition at involution in *Xenopus* (*Figures 1* and *6*) explains the dominance of the symmetric circumblastoporal forces of CT in the pre-involution region of the early gastrula, and the dominance of the asymmetric, CE forces in the post-involution region of *Xenopus* embryos beginning from the midgastrula stage (*Keller and Danilchik, 1988*). At the other end of the spectrum of CT-CE transition, *Gastrotheca riobambae* delays all CE until neurulation and has a completely symmetrical blastopore closure, both externally and internally (*del Pino, 1996*). Others, such as the direct developing *Eleutherodactylus coqui* and *Epipedobates tricolor*, show intermediate CT-CE transitions (D. Shook, personal observations). In ongoing work, we are testing the idea that the deployment of CT and CE varies with the egg size and the amount and distribution of yolk and the mechanical challenges changes in these parameters present to the morphogenic machines of gastrulation.

## Conclusions

Our findings illustrate that CT is one of the morphogenic machines that contribute to blastopore closure, along with CE and Vegetal Rotation (*Winklbauer and Schürfeld, 1999*) and that CT is capable of closing the blastopore on its own in the absence of CE patterning, as in ventralized embryos. If we assume that the cell motility driving CT is not dependent on the PCP pathway, since it does not depend on polarized motility, it appears that CT clearly can not reliably close the blastopore in the presence of patterned but non-functional CE (*Djiane et al., 2000*; *Tada and Smith, 2000*; *Wallingford et al., 2000*; *Habas et al., 2001*; *Goto and Keller, 2002*; *Habas et al., 2003*; *Ewald et al., 2004*), a hypothesis we test elsewhere (*Shook et al., 2018*). Presumably, blocking CT while allowing CE to occur in an inappropriate context would also block blastopore closure. Normal blastopore closure is the result of the coordinated expression of this system of machines in an appropriately configured biomechanical context, and is thus a problem in systems biology. Our demonstration of the role of CT in this system furthers our understanding and ability to study this system.

Failure of amphibian blastopore closure is not a 'non-specific phenotype', but results from a failure of some part of this system, which as we have shown here, can be teased apart biomechanically.

## Methods

### Embryo culture, manipulation and explant construction

*X. laevis* embryos were obtained and cultured by standard methods (*Kay and Peng, 1991*), staged according to Nieuwkoop and Faber (*Nieuwkoop and Faber, 1967*), and cultured in 1/3X MBS (Modified Barth's Saline). For explants made before stage 10 the embryos were tipped and marked to identify the dorsal side (*Sive et al., 2000*). Standard 'giant' sandwich explants were made at stage 10 to 10.25 as described previously (*Shook et al., 2004*)(see also *Figure 2—figure supplement 2A*), and modifications for Dorsal 180°, and Ventral 180° explants are described here (*Figure 2A–E*). Explants are cultured in Danilchik's for Amy (DFA) (*Sater et al., 1993*). Ventralized giant explants were made from ventralized embryos (see below) (*Figure 2F,G*) without reference to 'dorsal' as these embryos are symmetrical about the blastopore (*Scharf and Gerhart, 1980*)(*Video 1*). 'Unencumbered' explants were those without a load (e.g. unrestrained in the measuring apparatus, below). For explants made before bottle cells had formed, the vegetal endoderm was cut away from the circumblastoporal region just below the transition in cell size, above which most bottle cells will form (*Keller, 1981*). Explants were staged by time elapsed from stage 10 control embryos, and when the dorsal bottle cells began to re-spread (stage 11) (*Hardin and Keller, 1988*). Animal cap sandwiches were made from the ventral portion of the blastocoel roof of stage 10 embryos (not shown).

### Ventralization of embryos

De-jellied embryos were placed in dishes made of 15 mm transverse sections of 60 mm diameter PVC pipe with Saran wrap stretched across the bottom, irradiated 6 or 7 min from below at about 35 min post fertilization on a UV trans-illuminator (analytical setting (Fotodyne Inc. Model 3–3500) and left undisturbed for at least an hour to avoid accidentally rotating and thus dorsalizing them (*Black and Gerhart, 1986*). Embryos were also ventralized by injecting β-catenin morpholino vegetally into the first two blastomeres (*Heasman et al., 2000*). Embryos forming bottle cells asymmetrically or earlier than the majority of ventralized embryos were discarded as being insufficiently ventralized. Control, ventralized embryos were cultured to control stage 35–38 and scored for their DAI (*Kao and Elinson, 1988*) or to control stage 28 and stained for dorsal markers (Appendix 2) to evaluate the effectiveness of ventralization.

### Image analysis

Explant morphometrics (see Appendix 3) were done with NIH *Figure 1.6* software (Wayne Rasband, National Institutes of Health; available at http://rsb.info.nih.gov/nih-image/; RRID:SCR_003073), Object Image (Norbert Vischer, University of Amsterdam; available at https://sils.fnwi.uva.nl/bcb/Object-Image/object-image.html; RRID:SCR_015720) or Image J (http://rsb.info.nih.gov/ij/; RRID:SCR_003070).

### 'Tractor pull' biomechanical measurement apparatus

Explants were attached to two polyester shim stock strips (Small Parts, Inc. cat. # SHSP-010), both 25 µm thick x 0.8 to 1.5 mm wide, and one, the 'anchor', 4 to 8 mm long, and the other, the 'sled', 3 to 5 mm long (*Figure 2H*). A cleat of shim stock (Small Parts, Inc. cat# SHSP-200), 500 µm thick x ~ 500 µm on a side, was glued to the sled with clear fingernail polish (Sally Hansen 'Hard as Nails') (*Figure 2H*). The anchor and sled were coated with fibronectin (Roche cat # 11 080 938 00, at 20 µg/ml, in 1/3X MBS for 30–60 min at 37°C) and inserted 0.5 to 1.0 mm (15–30% of the mediolateral extent of the explant) between the inner faces of the lateral ends of the sandwich to allow attachment (30–60 min) (*Figure 2H*). The explant was placed over a window of cover glass (#1.5) in a 100 mm culture dish, and the anchor was attached to the window with high vacuum silicone grease (Dow Corning, Inc.) (*Figure 2I*). Using the stage controls, the cleat on the sled was moved adjacent to a calibrated optical fiber probe (40–50 mm by 120 µm, Mouser, Inc. stock 571–5020821) mounted on the end of an aluminum rod fixed to an XYZ micromanipulator attached to an IX70 Olympus inverted

microscope. The spring constants of these cantilever probes were calibrated by measuring deflection upon hanging short lengths of wire of known length/mass on a reference probe, which was then used to calibrate other probes. Five probes were made; the first two were discarded because of damage or unsuitability, probes #3 and #4 were used for measurements here, while probe #5 was used as the reference probe. Tension on the explant was measured by probe deflection recorded in high-resolution movies from below the window (40x objective, Olympus IZ70) and its behavior recorded simultaneously by time-lapse imaging from above (Olympus stereoscope). Glass beads (106 µm diameter, Sigma Cat#G-4649) between the sled and the window and a 200 to 300 µm thick, 1% agarose bed between the explant and the window lowered friction (*Figure 2I*). (note: since doing these experiments, we have learned that beads of about 250 µm diameter give lower friction). Probe drift and sled-substrate friction were characterized (Appendix 1).

## Force measurement test

Mediolateral tensile force was measured as the explants pulled the cleat of the sled against the probe. In most cases, the probe was placed adjacent to the cleat such that both probe and explant were unloaded at the start of the experiment. In others, the explant was 'pre-strained' about 25%, by moving the cleat against the probe, and then away from it, or the explant was left under a pre-tension. Measurements were made to tailbud stages (~20 hr) when the probe deflection generally ceased to increase significantly. The cleat was then backed off from the probe to determine the resting position of the probe. Tension could be adjusted during a force measurement by moving the anchor away from or toward the sled, to decrease or increase strain (as in *Figure 4A–D*).

Probe position was recorded every six minutes, and probe displacement was translated into force by the following:

$$F_{(1)} = D_{(t)} \bullet M \bullet K_P \tag{1}$$

where D is displacement in pixels, M is the magnification scale in µm/pixel and $K_P$ is the spring constant of the probe in µN/µm. Drift was determined from the unstressed position of the probe before and after the assay, the difference interpolated linearly over the duration of the test and subtracted from the probe movement. Means of force traces were plotted, with the mean of hourly intervals and the standard error of the hourly mean shown as error bars.

## Structural tensile stiffness measurement: Uniaxial tensile stress relaxation test

For estimates of stiffness (*Wiebe and Brodland, 2005*), anchor-explant-sled preparations were mounted as for force tests, and strain was applied along their mediolateral (circumblastoporal) axis by moving the stage 300 µm (10–12% strain) over one to a few seconds. Relaxation was allowed for 5 min, the stage was withdrawn 400 µm from the probe, and any further shape change were recorded (recovery). Probe positions were recorded every 0.5 to 30 s, and the anchor-explant-sled assembly was imaged every 1–10 s. The stage position was determined by a calibrated Metamorph image processor. The strain imposed on the explant was based on the relative position of two points in the explant that lay above the medial edges of the attachment strips. The explants were tested periodically during gastrulation and neurulation with unstressed periods (>1 hr) between tests.

We modeled the time (t) dependent structural stiffness (SS(t), Pascals) using the following viscoelastic spring and dashpot model (*Findley et al., 1989*; *Moore et al., 1995b*)(Appendix 4):

$$SS(t) = S_{inf} + S_{sp} \bullet e^{(-t/\tau)} \tag{2}$$

where the parameters are $S_{inf}$ or stiffness at infinite time (residual stiffness), $S_{sp}$ or instantaneous stiffness, and $\tau$, the relaxation time constant, representing the half life of stress-relaxation. $\eta$, the coefficient of viscosity, is related to $\tau$ by:

$$\eta = \tau \bullet S_{inf}$$

The model assumes that instantaneous stiffness is reduced by viscous flow or remodeling of intra- or inter-cellular structures, until a residual stiffness, representing stable elastic elements of intra- or inter-cellular structure, is reached. SS(t) was calculated using the cross-sectional area of the tissue,

force measurement, and observed strain over time, and two analytical techniques were used to generate the parameters $S_{inf}$, $S_{sp}$ and $\tau$ (Appendix 5). Alternatively, an explant 'spring stiffness' constant ($K_E$) was also calculated. See Appendix 4 for further details.

## Acknowledgements

We dedicate this work to the memory of two Professors Emeritus, Antone G. Jacobson (1929–2017), University of Texas, Austin, a true pioneer in the biomechanical analysis of morphogenesis, notably its cellular basis and mechanical coupling between tissues (*Jacobson and Gordon, 1976*), and to the memory of Professor Lev V. Beloussov (1935–2017), Lomonosov Moscow State University, Moscow, whose insightful analysis of mechanical stresses in frog embryos (*Beloussov et al., 1975*) inspired this work. We thank Rudi Winklbauer and three anonymous reviewers for helpful suggestions that improved the manuscript.

## Additional information

### Funding

| Funder | Grant reference number | Author |
| --- | --- | --- |
| National Institutes of Health | NICHD R37 HD025594 MERIT AWARD | Raymond Keller |

The funders had no role in study design, data collection and interpretation, or the decision to submit the work for publication.

### Author contributions

David R Shook, Conceptualization, Data curation, Formal analysis, Supervision, Investigation, Methodology, Writing—original draft, Project administration, Writing—review and editing; Eric M Kasprowicz, Investigation, Writing—review and editing; Lance A Davidson, Conceptualization, Methodology, Writing—review and editing; Raymond Keller, Supervision, Funding acquisition, Investigation, Project administration, Writing—review and editing

### Author ORCIDs

David R Shook (iD) http://orcid.org/0000-0002-0131-1834
Lance A Davidson (iD) http://orcid.org/0000-0002-2956-0437

### Ethics

Animal experimentation: This study was performed in strict accordance with the recommendations in the 8th Edition of the Guide for the Care and Use of Laboratory Animals, of the National Institutes of Health. All of the animals were manipulated according to an approved institutional animal care and use committee (IACUC) protocols of the University of Virginia. The protocol was approved by the Animal Care and Use Committee of the University of Virginia (protocol #2581). All surgery was performed under Tricaine anesthesia, and every effort was made to minimize suffering. The animal care and use program is accredited by the Association for Assessment and Accreditation of Laboratory Animal Care, International (Date of most recent AAALAC accreditation: 11-22-2016). The University of Virginia has a PHS Assurance on file with the Office of Laboratory Animal Welfare (OLAW) (PHS Assurance #A3245-01, Valid through 06-30-2019 ). The University of Virginia is a USDA registered research facility(USDA Registration # 52-R-0011, Valid through 08-22-2017).

### Decision letter and Author response

Decision letter https://doi.org/10.7554/eLife.26944.042
Author response https://doi.org/10.7554/eLife.26944.043

## Additional files

### Supplementary files

• Transparent reporting form
DOI: https://doi.org/10.7554/eLife.26944.032

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

## Appendix 1

DOI: https://doi.org/10.7554/eLife.26944.033

## Drift and friction:

Probes were placed in distilled water and imaged every 3 min. Force equivalents for the observed drift were calculated, for purposes of comparing different probes and the effect of drift on force measurements. Probe #3 showed minimal drift, within ±0.5µN over 6 hr (*Figure 3—figure supplement 2A*). In most cases the most rapid drift occurred within the first 60 min. Probe #4 shows much greater drift (*Figure 3—figure supplement 2B*), especially when recorded from shortly after the dry probe was immersed (*Figure 3—figure supplement 2C*). However, in nearly all cases, within 30 min of immersion probe #4 showed drift that was near linear over the range of forces we consider. When probe #4 was allowed to soak overnight, it had reduced probe drift (turquoise line that reaches 0.5µN, *Figure 3—figure supplement 2B*). Drift for probe #4 may have to do with hydration and/or temperature equilibration of some element of the probe-holder assembly. By soaking the probes for 60 min prior to use and correcting for any remaining drift (as measured at the end of the tractor pull, upon the release of the explant from the probe) by interpolating over the period of force measurement, we minimized the error contributed by drift.

To evaluate the friction between the sled and the under-laying substrate, we moved the stage holding the tractor pull chamber such that the cleat on an unattached sled was pushed against a stationary probe, recorded the probe position every 100–200 s and calculated the force exerted on the probe at each time for 2–6 hr. We compared the friction of the sled with the substrate, (1) on bare glass, 2) with an agarose pad of roughly 50–100% larger area under the sled and (3) with a sparse layer of 100 µm glass beads under the sled. Tests were done in DFA, which contains 0.1% BSA. A representative example of test runs on different substrates is shown (*Figure 3—figure supplement 3*). Treatment of the glass beforehand (e.g. by acid-ethanol wash and/or coating with BSA) appeared to have minimal effect. For glass, agarose and beads (n's = 4, 5 and 7), the average median forces were 0.41, 0.48 and 0.25 µN respectively. The average minimum forces were 0.27, 0.36 and 0.00 µN. The average standard deviation was 0.10, 0.07 and 0.14 µN. The force on the probe was below 0.32, 0.40, and 0.09 µN 10% of the time. And the fraction of the time forces were below 0.25 µN was 12, 23% and 64%. Although the 1.5 to 2-fold differences in the median friction force was only a moderate improvement, the difference in the time the friction force was at or near 0 µN on beads was dramatic. Rather than experiencing a fairly steady approximately 0.4 to 0.5 µN of friction in the case of glass or agarose, sleds over beads experienced lower friction much more frequently. For this reason, we used a sparse layer of beads in all force measurement and stress-relaxation tests. We assume that force measurements are approximately 0.2 µN below the force explants could produce at a given time.

## Appendix 2

DOI: https://doi.org/10.7554/eLife.26944.034

### Immunohistochemistry and ventralized embryos.

For immunohistochemistry, embryos and explants were fixed at stage 26–28 in MEMFA (*Kay and Peng, 1991*) overnight at 4°C and transferred to methanol for storage at –20°C. Fluorescent staining for notochord with Tor70 (temporary Antibody ID is AB_2715462, *Kushner, 1984*) and for somitic mesoderm with 12/101 (RRID:AB_531892, *Kintner and Brockes, 1984*) was done as previously described (*Bolce et al., 1992*). Notochord and somites in giants from tractor pulls were generally elongated orthogonal to the mediolateral axis, although to a lesser extent than in an unencumbered giant. In standard giant or D180° sandwich explants from tractor pulls, the posterior ends of the two notochords frequently did not fuse, and in some cases most of the notochords were separate, but co-linear (*Figure 2—figure supplement 4A–H*). The two files of somites do fuse (*Figure 2—figure supplement 4A–H*). In tractor pull explants where an additional strain was imposed, both posterior notochord and somites often did not fuse, and sometimes elongate non-orthogonally to the axis of pull (*Figure 2—figure supplement 4I–L*); in a few cases, the two sets of notochords and somites remained largely independent, which tended to be coupled with generally aberrant morphogenesis.

UV ventralization for 5 to 7 min gave an average DAI of 1.8 (n = 318). Embryos of DAI score 0 to 3 generally manifested little or no evidence of CE prior to the end of neurulation, indicating that we had effectively eliminated CE in our embryos during the period of force measurement. Ventralized sandwich explants in some cases contain small amounts of somitic tissue, but rarely show any notochordal tissue (*Figure 2—figure supplement 4M–P*). Among unselected embryos, some notochord appeared infrequently (4%, n = 56) while some somite appeared more frequently (40%, n = 54). When ventralized giant sandwich explants did have some dorsal tissue, it generally didn't manifest (show any sign of CE) until after the plateau had been reached, and generally detracted rather than added to the force.

## Appendix 3

DOI: https://doi.org/10.7554/eLife.26944.035

### Morphometrics.

The sagittal sectional area of unconstrained giants sandwiches was determined from RDA-labeled explants (50 ng/embryo), cultured to control stages 10.5 to 19, fixed in MEMFA and imaged in the laser scanning confocal. Minimal changes in explant dimensions were seen after fixation (<5%, n = 6). The Z-step distance of the confocal was calibrated using a coverslip fragment of known thickness immersed in a solution of RDA. En face confocal images for the entire explant were obtained, re-sliced to show the mid-sagittal sectional plane, and the area of these plotted against stage. A regression was plotted on the sagittal sectional area (SSA) of several explants, and this was used to estimate the sagittal sectional area of the explants used in the stress-relaxation test. To estimate the effective SSA, explants were sliced parasagittally and parasagittal confocal images were collected, from which the area of the deep mesoderm was estimated, constrained by cell size and distance from the bottle cells.

Proportional convergence or strain in sandwich explants along the entire mediolateral axis of the IMZ or for defined sub regions was measured with respect to the distance between specific cells or distinctively pigmented regions along the widest part of the IMZ (generally, near the limit of involution) at the onset of time-lapse recording ($L_{(i)}$), and thereafter ($L_{(t)}$). Time specific strain was calculated as:

$$S_{(t)} = (L_{(t)} - L_{(i)})/L_{(i)}$$

Rates were then $S_{(t)}/\Delta t$. Convergence is expressed as strain $* -1$. Rates of convergence during giant construction were instead with respect to the initial circumference in the intact embryo.

Shear rate of explant with respect to attachment strips for a given time period was calculated as:

Shear rate = $(\Delta W - \Delta D)/W(i)/\Delta t * 100\%$

where W(i) is the initial width of the widest part of the mesendoderm at the onset of the assay, $\Delta W$ is the change in width during the time period, $\Delta D$ is the displacement of the sled strip toward the anchor strip during the time period and $\Delta t$ is the elapsed time.

## Appendix 4

DOI: https://doi.org/10.7554/eLife.26944.036

### Estimation of parameters

Two alternative methods were used to determine parameters for the spring and dashpot model

$$S_{(t)} = (L_{(t)} - L_{(i)})/L_{(i)}$$

based on the observed time (t, in seconds) dependent structural stiffness (SS), calculated as:

$SS_{(t)} = F_{(t)} /(SSA \cdot S_{(t)})$ (3)

where time specific force ($F_{(t)}$) was as calculated in **Equation 1** (main text), SSA was the estimated stage specific sagittal sectional area (see Appendix 3), and $S_{(t)}$ the time specific strain (see Appendix 3) was measured from the mediolateral extent of the mesodermal component of the explant spanning the gap between the strips to which they were attached.

In the first method ('log transform'), $SS_{INF}$ is assumed to be 0.97 * $SS_{(180)}$ because stress decay has stabilized by 180 s, and active convergence is likely to overwhelm further relaxation. Given **Equation 2**, a linear regression on

$y(t) = \ln(SS_{(t)} - SS_{INF})$

then yields the line

$y(t) = (-1/\tau) * t + \ln(SS_{SP})$ and $SS_{INF}/SS_{(180)} = \sim 0.97$

(see **Appendix 4—figure 1A**)

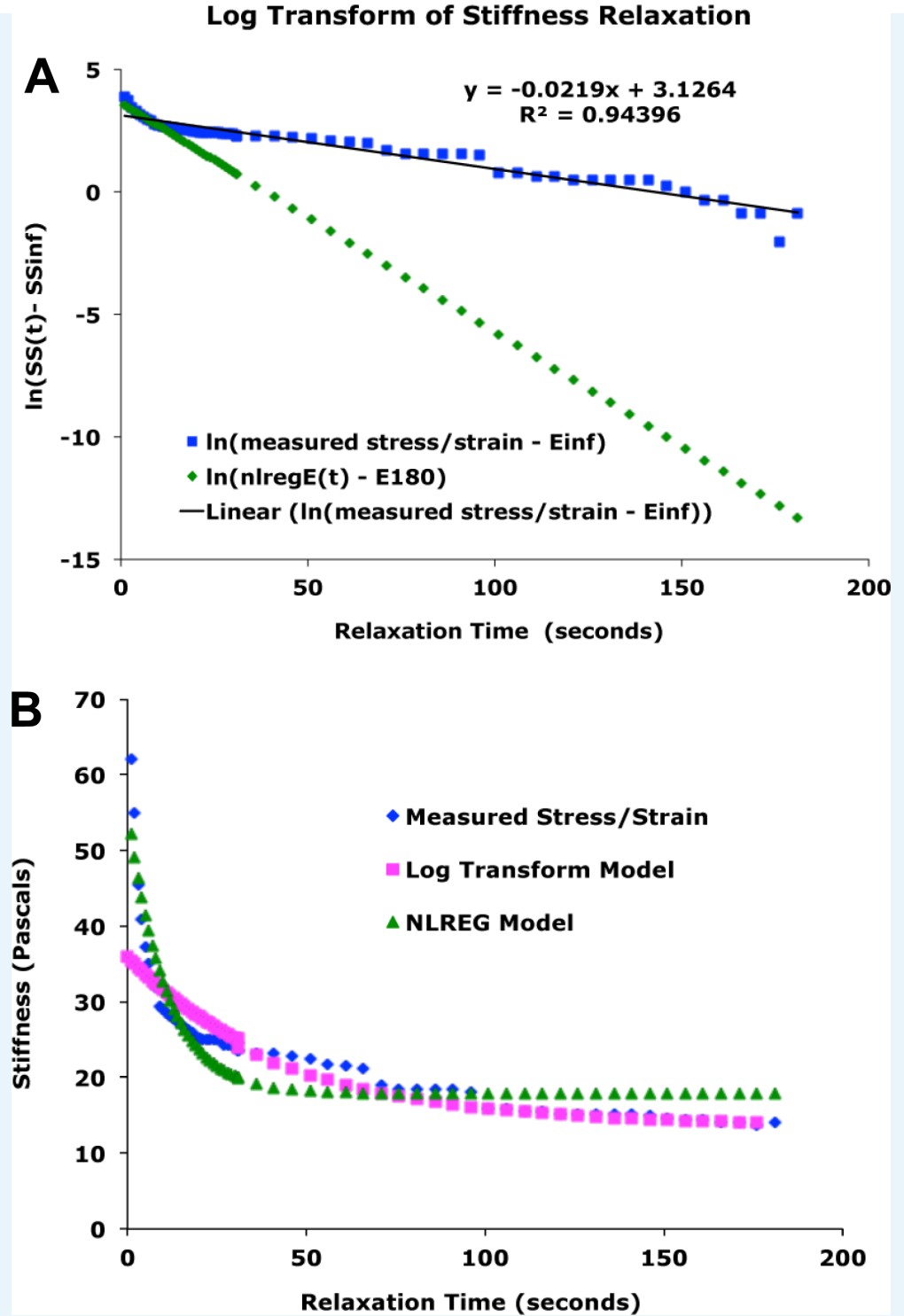

**Appendix 4—figure 1.** Comparison of methods for estimating parameters. (**A**) Example of estimation of parameters from linear regression on log transform of time-dependent stiffness data. Blue: plot of ln(measured SS(t) – measured SS(180) * 0.97)). Black: linear regression plot on log transformed data. green: plot of ln(SSsp * e(-t/tau) – E180), with parameters as estimated by NLREG. (**B**) Comparison of Model vs. Measured Stiffness for the case in A, above. Blue: measured time dependent structural stiffness (SS(t)). Magenta: plot of E(180) + SS(sp) * e$^{(-t/tau)}$, with the later two parameters based on linear regression on log transform, as in A, above. Green: plot with all parameters from NLREG.

DOI: https://doi.org/10.7554/eLife.26944.037
The following source data is available for figure :
**Appendix 4—figure 1—source data 1.** Source data for parameter estimation method examples.
DOI: https://doi.org/10.7554/eLife.26944.038

In the second method, a curve-fitting program (NLREG, RRID:SCR_015722, available at http://www.nlreg.com, *Sherrod, 1995*) was used to generate the parameters $SS_{INF}$, $SS_{SP}$ and $\tau$. NLREG tended to generate a higher $SS_{SP}$ and a correspondingly more rapid time relaxation constant (*Appendix 4—figure 1*).

In both cases, viscosity ($\eta$) is determined as:

$\eta = SS_{SP} * \tau$

Viscosity and instantaneous stiffness derived using the log transform method both show a significant increase between late gastrulation and mid-neurulation (*Figure 5—figure supplement 1A,B*, green), whereas using NLREG, they show no significant difference (*Figure 5—figure supplement 1A,B*, orange). These viscosity estimates are roughly an order of magnitude lower than those measured by *David et al., 2014* on deep tissue alone. The major difference in the current study is that explants are deep tissue enclosed in superficial epithelium, which should lower the tissue surface tension.

The log transform method tends to match the later part of the stress-relaxation curve, giving a lower instantaneous stiffness and higher viscosity, whereas the reverse is true of the NLREG method (*Appendix 4—figure 1B*). The two methods highlight the fact that viscosity appears to be much lower during the first 10–15 s of stress relaxations than thereafter. This may reflect a change in the cellular elements that are viscously flowing over time, with very low viscosity elements flowing first, followed by successively higher viscosity elements. Because tissues in the embryo are already tension bearing, the viscosity derived form log regression more accurately reflects the relevant mechanical properties over developmental time scales.

An explant 'spring stiffness' constant ($K_E$) was also calculated:

$$K_E = F_{(180)}/(L_{(180)} - L_{(0)})$$

with $F_{(t)}$ and $L_{(t)}$ as described above.

## Appendix 5

DOI: https://doi.org/10.7554/eLife.26944.039

### Caveats

Our stiffness measurements represent an approximation of composite structural stiffness, rather than the true stiffness of a uniform material. Giant sandwich explants are composed of presumptive endodermal and mesodermal tissues vegetally and ectodermal tissues animally, of different presumptive mesodermal (notochord, somite, etc.), ectodermal (epidermal, neural) and endodermal (vegetal, suprablastoporal) along both mediolateral and animal-vegetal axes, and different basic tissue types (epithelial, mesenchymal) along their radial dimension (perpendicular to the plane of the explant) so are not homogeneous in any dimension. Additionally, especially after CE begins at stage 10.5 (G+2hr), the shape of the sagittal sectional area (SSA) begins to vary along the mediolateral extent of giant sandwich explants (*Figure 2—figure supplement 3*; *Video 2*). As a consequence of these inhomogeneities, stiffness measures are biased toward the least stiff regions along the mediolateral axis.

In D180 and V180 explants at later stages, the mesodermal tissue rounded up to some extent, with a relatively circular cross section across the AP axis, with the attachment strips inserted part way into the circle. Stretch resulted in both over-all strain of the mesodermal tissue, but also flattening of the circular cross section. Probably as a consequence, later stiffness measurements from D180° explants in particular are more variable (*Figure 5C*). We found that straining these rounded tissues 600 rather than 300 microns gave more consistent results for stiffness measurements, and so those results are reported.

## Appendix 6

DOI: https://doi.org/10.7554/eLife.26944.040

### Supplementary discussion

The nature of convergence by MIB may offer an alternative or complementary explanation as to why an increase in compressive load results in increased force whereas an increase in tensile load does not. The cell intercalation process is self-reinforcing in that it increases the number of units in the parallel, pulling array, but depends on the generation of additional tension in order for cells to pull themselves in between one-another. Thus some threshold of unresolved tensile load may retard further convergence via intercalation and thus limit additional force production, whereas an increased compressive load driving extension, the output of convergence, may activate compression-sensitive accommodation mechanisms while not immediately limiting the progress of intercalation. The differences in responses to compressive and tensile loads should be evaluated further.

