## [Decision Letter]

Thank you for submitting your article "Large, long range tensile forces drive convergence during *Xenopus* blastopore closure and body axis elongation" for consideration by *eLife*. Your article has been reviewed by three peer reviewers, and the evaluation has been overseen by a Reviewing Editor and Didier Stainier as the Senior Editor. The reviewers have opted to remain anonymous.

The reviewers have discussed the reviews with one another and the Reviewing Editor has drafted this decision to help you prepare a revised submission.

Summary:

In this manuscript the authors clarify the specific biomechanical role of convergent thickening and convergent extension that impacts *Xenopus* embryo morphogenesis during gastrulation. The results of the work address important, long-standing questions in the field regarding tissue mechanics of vertebrate development and the studies appear to be rigorously performed and analyzed. Overall the work is of high quality and should represent an important contribution to the literature provided that most of the suggested revisions by the reviewers can be addressed.

Essential revisions:

Much of the writing and presentation of the results is hard for non-specialized researchers to easily understand, thus revising the text throughout, and particularly the Introduction and Discussion, is needed to make the fundamental concepts more readily accessible and comprehensible. Consultation with a non-expert in the details of this work could help to make suggestions to the figures and text, or at least specifically point out elements that could benefit from some such clarification.

Validation and/or application of the proposed model should be performed to demonstrate its relevance to tissue morphogenesis.

The rationale for, and implications of, the stiffness calculations for CT/CE-driven morphogenesis should be further clarified.

*Reviewer #1:*

In this manuscript, the authors define the biomechanical basis of the convergence (tissue shortening) forces that drive blastopore closure and extension of the body axis in *Xenopus* embryos. They focus on two different morphogenetic machines believed to drive this convergence in the involuting marginal zone (IMZ) tissue of the gastrula: convergent thickening (CT) and convergent extension (CE).

Using a custom-built mechanical measuring device (a "tractor pull" apparatus, previously reported elsewhere) to directly quantify the forces generated by IMZ explants, the authors show that, during early gastrulation, CT generates circumblastoporal tension throughout the (pre-involution) IMZ. They then identify a transition point at mid-gastrulation, after which CE contributes progressively more force (via post-involution mediolateral intercalation behavior).

The manuscript also describes stress-relaxation tests to calculate the tensile stiffness of IMZ tissues (by modeling IMZ explants as viscoelastic materials). From these endeavors, the authors conclude that all tissues of the embryo gradually increase in stiffness by the end of gastrulation.

Previously, the relative contribution of CT vs. CE in convergence was not known, and CT was thought to be limited to only the ventral sector of the gastrula. The data presented in this manuscript clarifies the independent spatiotemporal influences of CT and CE on convergence, and quantifies the force dynamics involved at different stages of development. The authors conclude with an elegant model of the function of CT and CE as integrated spring (CE) and motor (CT) pulling mechanisms, offering a deeper, biomechanical understanding of the convergence forces likely acting in vivo during *Xenopus* gastrulation.

1) It is not entirely clear (to me) how the tensile stiffness calculations are related to the force measurements, nor how stiffness was used to inform the model in Figure 6. The rationale for, and implications of, the stiffness calculations for CT/CE-driven morphogenesis should be further clarified.

2) The conclusions regarding the timing and independence of CT vs. CE are interesting and could provide a reinterpretation and refinement of our understanding of the biomechanical mechanisms driving gastrulation. However, it is difficult to predict the impact and significance of this work without some sort of validation or application of the proposed model. For example, the tractor pull apparatus could be used to biomechanically tease apart a "non-specific" blastopore closure/gastrulation phenotype (e.g., elicited by PCP gene misexpression), thus testing the validity of the model and demonstrating its relevance for understanding morphogenesis.

*Reviewer #2:*

This paper is a welcome addition to the canon of *Xenopus* gastrulation literature on three levels. First, it provides new data on the patterning of forces during gastrulation in one of the key model system for understanding this important developmental process. This is important, as the key role of mechanics is now well appreciated, but we still lack solid data on many aspects of gastrulation in vertebrates.

Second, it brings to the fore the previously under-appreciated process of convergent thickening, which relies on radial intercalation rather than mediolateral cell intercalation. Indeed, we are seeing in the last year something of a renaissance of interest in radial intercalation after over a decade of obsession specifically with the mediolateral variety. So this aspect sis quite timely.

Third, through the first two, the paper provides answers to several longstanding puzzles in the field, including but not limited to: why do ventralized *Xenopus* embryos close their blastopores and how do other frogs – that express convergent extension only very late – close their blastopores.

Thus, the paper has impact on developmental and evolutionary issues, as well tissue mechanics.

The data appear solid throughout (I say "appear" only because I am not an expert in mechanics). And the while the writing is dense, so too are the data, so it seems reasonable here.

*Reviewer #3:*

The manuscript "Large, long range tensile forces drive convergence during *Xenopus* blastopore closure and body axis elongation" by Shook et al., provides direct evidence that two morphogenetic movements, convergent thickening (CT) and convergent extension (CE) of the marginal zone (MZ), generate tensile convergence forces to close the blastopore and elongate the body axis in *Xenopus*. The authors, measured these forces in several MZ explants (giants, dorsal 180°, ventral 180° and ventralized giants) that undergo CT and/or CE using a customized "Tractor Pull" device. The authors show that the tension developed by giant explants is increasing in two distinct phases separated by a plateau phase. Interestingly, the first phase of force increase is conserved in explants that do not undergo CE (ventral 180° and ventralized explants), suggesting that during this period (onset of gastrulation) CT is the main source of the circumblastoporal convergence tension, providing thus new insights into the yet non-well defined role of CT in *Xenopus* development. The second phase of force increase is at late neurulation which mainly coincides with CE behavior and stiffening of these tissues. Overall, this work confirms the long-standing notion that convergence tensile forces close the blastopore in amphibians. The finding that CT plays a major role in this process is interesting especially to the researchers of *Xenopus* developmental biology, since it is a morphogenetic movement that was not well-described before and it could provide a new avenue of interpreting phenotypes related to blastopore closure. The experiments are technically well performed and describe nicely the mechanical input of the two morphogenetic events (CT and CE) in *Xenopus* morphogenesis. However, the following major concerns need to be taken into consideration before publishing this study in *eLife*.

A) Although the authors mention that CT and CE are evolutionary conserved, in order for this study to be highly appreciated by a wider audience beyond the field of *Xenopus* developmental biology, it would be valuable to characterize at the cell and molecular level the differences between CT and CE (i.e. which cell behaviors and which molecules can polarize the cells to first converge and thick the tissue and then switch to elongation). Such mechanistic insight will be extremely interesting for the field of mechanobiology, since it will explain how different cell/molecular behaviors lead to distinct force generating processes that shape differently the tissue (please see below).

B) Although the authors provide several diagrams to explain to the reader the complexity of blastopore closure, the manuscript and figures are still very hard to be understood by researchers non-specialized in *Xenopus* gastrulation (for specific comments please see below).

Major points:

1) It was previously shown that inhibition of the PCP pathway in the dorsal MZ leads to inhibition of CE and defects in blastopore closure. In contrast, inhibition of the PCP pathway in the ventral MZ only leads to a slight delay of blastopore closure until mid-gastrula stages (until which CT is mainly occurring), indicating that CT might not rely on PCP signaling (Ewald, A.J. et al., 2004, Development). In order for the authors to provide cellular and molecular mechanistic insights, it would be worth to express Xdsh either dorsally or ventrally and examine if and how PCP signaling affects the convergence force generated by CT in all four types of explants, and how this can be correlated with defects in morphogenesis of the sibling embryos. Such experiments would not only support the impact of this study for the interpretation of the phenotypes associated with defective blastopore closure but will also provide evidence for differential molecular control of polarity during CT and CE (point A).

2) The giant explants exhibit two force increase phases separated by a plateau phase. Although the authors suggest that this could be associated with the progression of MIB, it would be relevant to show with high resolution time lapse imaging how cell behavior is changed in the different phases (i.e. cell polarity, protrusion formation) (point A).

3) As mentioned in the text, CT was previously described to take place at the ventral MZ and later in development (Keller, R., and Danilchik, M., 1988, Development), but here the authors show that it takes place in the whole MZ earlier in gastrulation. It would be useful to describe this finding further by showing images or time lapses of how CT progresses in both embryos and explants. Sagittal sections of ventral vs. dorsal explants will also be useful to address any differences in the timing and extent of CT in the two regions.

[Editors' note: further revisions were requested prior to acceptance, as described below.]

Thank you for resubmitting your work entitled "Large, long range tensile forces drive convergence during *Xenopus* blastopore closure and body axis elongation" for further consideration at *eLife*. Your revised article has been favorably evaluated by Didier Stainier (Senior Editor), a Reviewing Editor, and three reviewers.

The manuscript has been improved and it can be accepted pending the deposition of the other referenced manuscript (Shook et al., 2017) to bioRxiv, as previously discussed. Please work on this and resubmit when you're ready, citing the preprint and including a formal reference in the References list.

The full reviews are appended below for your consideration.

*Reviewer #1:*

This revised manuscript by Shook et al. investigates the biomechanics of convergent thickening (CT), a force-generating process previously implicated in blastopore closure during amphibian gastrulation. In this paper, the authors illuminate the nature of this process with quantitative biomechanical analyses of explanted amphibian embryonic tissues.

The reviews of the original manuscript asked authors to 1) revise the text throughout to provide clarification for non-specialists in the field, 2) include an experimental application/validation of the model, and 3) explain the rationale for, and implications of, the stiffness calculations. Below I address the perceived effectiveness of the revised manuscript in addressing each of these points.

1) The Introduction and Discussion of the revised manuscript are not any more streamlined than the original; however, the flow of information is more accessible to the non-specialist.

My main concern is that the new version of the Introduction now refers to unpublished data/conclusions about CT in an as-yet-unfinished additional manuscript (cited as Shook et al., 2017, although not included in reference list).

For example:

- "…our understanding of CT is that the underlying surface tension based mechanism does not require a polarized cell behavior, only motility to maximize high-affinity cell-cell contacts (Shook et al., 2017)." - "These events are based on…characterization of CT (Shook et al., 2017)." - "IMZ explants from ventralized embryos show a rapid, near uniform CT throughout the IMZ…(Shook et al., 2017)."

Thus, some of the background information about CT now included as the foundation of the present *eLife* manuscript (and presented in Figure 1) is somewhat misleading as, technically, it is not peer reviewed, published knowledge. In general, it is difficult to discern what is known (published), unknown (unpublished), and perhaps known only by the authors (but still unpublished) about CT as a morphogenetic machine.

2) Rather than include application/validation studies in the present manuscript, the authors chose to refer reviewers to independent data contained in the as-yet-unpublished, second manuscript (Shook et al., 2017).

Although the existence of this manuscript in preparation provides confidence that relevant experiments are ongoing and feasible, readers of the current *eLife* manuscript, when published, would not have access to these data. It does not seem prudent to allow the validity of the CT/CE model in the current *eLife* article to rely on findings that (while promising) have not yet been formally peer-reviewed/published.

Unfortunately, the current organization of all the relevant data regarding CT into two highly interrelated and interdependent studies, one of which is still unfinished, is simply less than ideal. (One would have hoped that the authors would have found a way to integrate some of the additional data into the present manuscript.)

3) The rationale for performing the stiffness calculations, and the implications of stiffness for CT/CE-driven morphogenesis, are much clearer.

*Reviewer #2:*

This revision addressed my previous concerns, all of which were minor. I strongly support publication.

*Reviewer #3:*

The manuscript has been revised along the lines suggested by the different referees. The revisions have substantially improved the manuscript – it is now much easier to read and the link to the other not yet published manuscript also helps in understanding the molecular and cellular basis of the described CT and CE mechanisms. That said, I now fully support publication of this manuscript as is.

---

## [Author Response]

Essential revisions:Much of the writing and presentation of the results is hard for non-specialized researchers to easily understand, thus revising the text throughout, and particularly the Introduction and Discussion, is needed to make the fundamental concepts more readily accessible and comprehensible. Consultation with a non-expert in the details of this work could help to make suggestions to the figures and text, or at least specifically point out elements that could benefit from some such clarification.

Based on consultation with non-experts, Figure 1 has been revised to more clearly illustrate the morphogenic mechanisms involved, as has the figure legend and text describing it in the Introduction. The relationship between CT and RI has also been clarified in the Introduction section; to be clear, they are to some extent opposite processes, driven by very different mechanisms, not two facets of the same process. The biomechanical measurements of stiffness have been better explained for understanding by non-specialists, as have their rationale and implications, in the Results and Discussion sections. The explanation of our model for the differential use of CT and CE (Figure 6) in the Discussion section has also been revised to clarify the model and its implications.

We welcome further suggestions from the reviewers for points that could be better clarified.

Validation and/or application of the proposed model should be performed to demonstrate its relevance to tissue morphogenesis.

As noted above, we have performed the experiments suggested by reviewers and include them in the CT paper, along with other that serve to validate our model and demonstrate its relevance to morphogenesis.

The rationale for, and implications of, the stiffness calculations for CT/CE-driven morphogenesis should be further clarified.

As noted above, we have included explanatory text for non-specialists, both more clearly explaining what we are measuring, why we are measuring it, and how we measure it, particularly in the portion of the Results section describing these experiments, and in the Discussion with our interpretation of these results.

Reviewer #1:[…] 1) It is not entirely clear (to me) how the tensile stiffness calculations are related to the force measurements, nor how stiffness was used to inform the model in Figure 6. The rationale for, and implications of, the stiffness calculations for CT/CE-driven morphogenesis should be further clarified.

An overview of force and stiffness measurements in “lay-biologists” terms are now given at the beginning of the relevant sections within the Results, before going into more specific detail. The reason for making stiffness measurements, and their implications are now spelled out at the beginning and ends of the relevant sections.

Our model in Figure 6 is supported by our stiffness measurements in that the differential “spring stiffness” of dorsal vs. ventrolateral tissues is used to explain the plateau, and the increasing spring stiffness of standard giants and dorsal 180s, but not ventral 180s, between about 9 and 12 hours, is used to explain the end of the plateau, with the eventual transition from CT to MIB in the ventral tissues of a standard giant.

The section in the Discussion section titled “The force profile reflects the transition from CT to CE and illuminates the mechanics of gastrulation”, as well as minor elements of Figure 6 have been changed to better explain the implications of our understanding of the forces produced by and the stiffness of embryonic tissues, both for the behavior of explants in the tractor pull and in the intact embryo.

2) The conclusions regarding the timing and independence of CT vs. CE are interesting and could provide a reinterpretation and refinement of our understanding of the biomechanical mechanisms driving gastrulation. However, it is difficult to predict the impact and significance of this work without some sort of validation or application of the proposed model. For example, the tractor pull apparatus could be used to biomechanically tease apart a "non-specific" blastopore closure/gastrulation phenotype (e.g., elicited by PCP gene misexpression), thus testing the validity of the model and demonstrating its relevance for understanding morphogenesis.

Experiments validating our model are now in the CT paper, as described above. MIB-driven CE clearly has a much greater dependence on PCP signaling than does CT. Further, CE is also more dependent on Myosin II B.

Reviewer #2:[…] The data appear solid throughout (I say "appear" only because I am not an expert in mechanics). And the while the writing is dense, so too are the data, so it seems reasonable here.

We thank the reviewer for these comments.

Reviewer #3:[…] A) Although the authors mention that CT and CE are evolutionary conserved, in order for this study to be highly appreciated by a wider audience beyond the field of Xenopus developmental biology, it would be valuable to characterize at the cell and molecular level the differences between CT and CE (i.e. which cell behaviors and which molecules can polarize the cells to first converge and thick the tissue and then switch to elongation). Such mechanistic insight will be extremely interesting for the field of mechanobiology, since it will explain how different cell/molecular behaviors lead to distinct force generating processes that shape differently the tissue (please see below).

We agree, and have begun this analysis in the CT paper. And as soon as Dr. Shook is able to obtain funding, this work will be carried forward, in *Xenopus* as well as other amphibian species, and ultimately in invertebrate species; Hemichordates appear to use some variation of CT to close their blastopore.

B) Although the authors provide several diagrams to explain to the reader the complexity of blastopore closure, the manuscript and figures are still very hard to be understood by researchers non-specialized in Xenopus gastrulation (for specific comments please see below).

We hope our new and improved Figure 1 and our explanation for it in the text addresses this problem. It is worth noting that amphibian gastrulation involves a complex set of morphogenic mechanisms, which one research paper cannot hope to thoroughly explain. We have however tried to distill the most important elements of these mechanism and explain them as clearly as possible, and to explain their relevance to blastopore closure.

Major points:1) It was previously shown that inhibition of the PCP pathway in the dorsal MZ leads to inhibition of CE and defects in blastopore closure. In contrast, inhibition of the PCP pathway in the ventral MZ only leads to a slight delay of blastopore closure until mid-gastrula stages (until which CT is mainly occurring), indicating that CT might not rely on PCP signaling (Ewald, A.J. et al., 2004, Development). In order for the authors to provide cellular and molecular mechanistic insights, it would be worth to express Xdsh either dorsally or ventrally and examine if and how PCP signaling affects the convergence force generated by CT in all four types of explants, and how this can be correlated with defects in morphogenesis of the sibling embryos. Such experiments would not only support the impact of this study for the interpretation of the phenotypes associated with defective blastopore closure but will also provide evidence for differential molecular control of polarity during CT and CE (point A).

As noted above, this set of experiments has been carried out, and will be included in the CT paper. Comparing normal to ventralized embryos, Xdd1 overexpression strongly interferes with blastopore closure in normal embryos, which express both CE and CT, but not in ventralized embryos, which rely on only CT to close their blastopore. Similar results were found with force measurements when comparing the effects of Xdd1 on D180 vs. V180 sandwich explants; Xdd1 strongly inhibits force production in D180s, which rely on both CT and CE to generate force, but only weakly inhibits force production in V180s, which rely on only CT.

This is a somewhat cleaner experiment than injecting Xdd1 ventrally in normal embryos, since this tissue will eventually express MIB-driven CE.

2) The giant explants exhibit two force increase phases separated by a plateau phase. Although the authors suggest that this could be associated with the progression of MIB, it would be relevant to show with high resolution time lapse imaging how cell behavior is changed in the different phases (i.e. cell polarity, protrusion formation) (point A).

The normal progression of MIB in the IMZ has been extensively documented by the Keller lab, in references listed in the introduction at the end of the explanation for Figure 1, as well as the failure of MIB to advance as rapidly as normal when CE is mechanically retarded, as in the tractor pull (but using other mechanisms). While it would be nice to be able to directly correlate the cell behaviors that are happening in explants in the tractor pull with force generation, the experimental preparations that allow high resolution imaging of cell behaviors are incompatible with force measurements. Technological advances may some day allow this, but it is not currently possible.

3) As mentioned in the text, CT was previously described to take place at the ventral MZ and later in development (Keller, R., and Danilchik, M., 1988, Development), but here the authors show that it takes place in the whole MZ earlier in gastrulation. It would be useful to describe this finding further by showing images or time lapses of how CT progresses in both embryos and explants. Sagittal sections of ventral vs. dorsal explants will also be useful to address any differences in the timing and extent of CT in the two regions.

Video 2 (included in the original submission) shows CT in the whole IMZ during early gastrulation. This has now been explicitly pointed out in the text. Video 1 shows whole embryos, but this provides little clarity on its own. These issues are also extensively documented in the CT paper.

[Editors' note: further revisions were requested prior to acceptance, as described below.]

The manuscript has been improved and it can be accepted pending the deposition of the other referenced manuscript (Shook et al., 2017) to bioRxiv, as previously discussed. Please work on this and resubmit when you're ready, citing the preprint and including a formal reference in the References list.

We have submitted a more complete, polished version of the draft of our Convergent Thickening paper on BioRxiv and cited it as (Shook et al., 2018) (https://doi.org/10.1101/270892) within the current manuscript (the Tractor Pull paper). We also added some acknowledgements and added DOIs to some of the references. Otherwise, the Tractor Pull paper is unchanged.